# Genome structure-based Juglandaceae phylogenies contradict alignment-based phylogenies and substitution rates vary with DNA repair genes

Ya-Mei Ding[1,3], Xiao-Xu Pang[1,3], Yu Cao[1], Wei-Ping Zhang[1], Susanne S. Renner [2] ✉, Da-Yong Zhang [1] ✉ & Wei-Ning Bai [1] ✉

In lineages of allopolyploid origin, sets of homoeologous chromosomes may coexist that differ in gene content and syntenic structure. Presence or absence of genes and microsynteny along chromosomal blocks can serve to differentiate subgenomes and to infer phylogenies. We here apply genome-structural data to infer relationships in an ancient allopolyploid lineage, the walnut family (Juglandaceae), by using seven chromosome-level genomes, two of them newly assembled. Microsynteny and gene-content analyses yield identical topologies that place *Platycarya* with *Engelhardia* as did a 1980s morphological-cladistic study. DNA-alignment-based topologies here and in numerous earlier studies instead group *Platycarya* with *Carya* and *Juglans*, perhaps misled by past hybridization. All available data support a hybrid origin of Juglandaceae from extinct or unsampled progenitors nested within, or sister to, Myricaceae. *Rhoiptelea chiliantha*, sister to all other Juglandaceae, contains proportionally more DNA repair genes and appears to evolve at a rate 2.6- to 3.5-times slower than the remaining species.

The walnut family, Juglandaceae, comprises 63 species in eight genera (Fig. 1) and includes some of the World's commercially most valuable nut-producing crops, such as Persian walnut, Chinese Iron walnut (both in the genus *Juglans*), pecan, and hickory (genus *Carya*). Many species also are valuable timber trees. Because of their economic importance, several genomes have been assembled to study fungal resistance genes and genes related to fruit quality, and their analysis has revealed an ancient whole-genome duplication (WGD) in the family[1–5], termed juglandoid WGD[1]. In addition, comparisons of the genomes of *Juglans regia* and *Carya illinoinensis* show biased fractionation and asymmetric loss of duplicated genes between two parental subgenomes that persist in these species[3,5,6], pointing to an allotetraploid origin of the family. This was also suggested in cytogenetic studies that explained the predominant Juglandaceae chromosome numbers of $2n = 32$ or 64 by assuming an ancestor shared with, or embedded in, Myricaceae, which have a basic number of $x = 8$[7,8].

Despite numerous molecular studies over the past 20 years, the relationships among the family's generally recognized genera have remained unclear, with at least six incongruent topologies having been reported[9–19], although the main signal in many of these studies came from the same chloroplast *matK* gene sequences. Most problematic has been the position of the enigmatic east Asian endemic *Platycarya strobilacea*, which has 'jumped' between major clades. Today, this species occurs in Vietnam, Taiwan Island, Mainland China, Korea, and Japan, but fossil *Platycarya* leaves and fruits are known from the Upper Paleocene and Early Eocene of North America and Europe[20,21].

[1]State Key Laboratory of Earth Surface Processes and Resource Ecology and Ministry of Education Key Laboratory for Biodiversity Science and Ecological Engineering, College of Life Sciences, Beijing Normal University, 100875 Beijing, China. [2]Department of Biology, Washington University, Saint Louis, MO 63130, USA. [3]These authors contributed equally: Ya-Mei Ding, Xiao-Xu Pang. ✉e-mail: srenner@wustl.edu; zhangdy@bnu.edu.cn; baiwn@bnu.edu.cn

*Platycarya* is unusual among the Juglandaceae in its cone-like fruit (Manning[22]; our Fig. 1), but the fossil leaves of *P. americana* and *P. castaneopsis* share unique leaf architectural traits with modern Engelhardieae[20], and extant *Platycarya* and *Engelhardia* further share thin nut walls without lacunae[22], terminal panicles in which the male and female flowers usually are combined (Stone[23]: Fig. 7.1), and similar pollen (Stone[23]: Fig. 7.3). At least four modern morphological data matrices[9,14,19,24] have been unable to unambiguously place *Platycarya*, even when fossil taxa were included (*Discussion*). The first such analysis[20], however, inferred a *Platycarya*/*Engelhardia* clade (also see Manchester[25]; our Fig. 1a).

Other genomic and population-genetic studies have highlighted the role of ancient and ongoing hybridization in the Juglandaceae's most diverse genus, *Juglans*[26,27]. If cross-species gene flow (introgression) has influenced large portions of related genomes, phylogenetic inference based on DNA-sequence alignments will produce species trees that reflect the reticulation history rather than the bifurcation history[28,29]. In such situations of extensive introgression, microsynteny appears to retain phylogenetic signal[30].

The molecular-clock hypothesis plays a central role in the study of molecular evolution, timing of speciation events, and historical demography. However, substitution rates can vary greatly among lineages, and gene duplicates can show distinct patterns of temporal and functional evolution[31], related to generation times[32], other life history traits (e.g., Lanfear et al. [33]), different population sizes[34–38], or possibly DNA repair systems[34,39,40]. This last possibility has received the least attention so far due to the technical difficulty of comparing the numbers of genes involved in DNA repair in different taxa.

Here, we use syntenic information and gene presence/absence to reconstruct the phylogenetic history of Juglandaceae, taking advantage of its two coexisting homoeologous subgenomes and chromosome-level genome assemblies. Our gene-content- and microsynteny-based phylogenetic approach builds on work by Zhao et al. [30] and Pett et al. [41], but differs from their studies by focusing on an allotetraploid clade. Resolving the phylogeny of the walnut family is important to better leverage the available genomes by knowing the direction of trait evolution in the family and because a correctly rooted topology is essential for biogeographic and molecular-clock studies. We also infer DNA substitution rates, past population sizes, and relative numbers of DNA repair genes to test the possibility that substitution rates may vary with the expansion of DNA repair gene families. In the context of other recent studies, our results highlight a pattern of dramatic post-polyploid evolutionary slowdowns in woody clades of core eudicots, calling for greater caution in molecular-clock dating of such clades.

## Results

### Genome assemblies and annotations

The total assembled genome of *Rhoiptelea chiliantha* comprised 408.19 Mbp, and the contig N50 and scaffold N50 values were 6.97 Mbp and 24.34 Mbp, respectively. Of the whole genome, 91.71% were ordered and oriented into 16 pseudo-chromosomes (Supplementary Fig. 1). The *R. chiliantha* genome consisted of ~39.35% repetitive sequences, and 32,505 predicted protein-coding genes (Supplementary Table 1). The *Engelhardia roxburghiana* assembly size was 884.78 Mbp, and 97.73% of the whole genome were ordered and oriented into 16 pseudo-chromosomes (Supplementary Fig. 1). The *E. roxburghiana* genome consisted of ~57.11% repetitive sequences and 30,590 annotated protein-coding genes (Supplementary Table 1). Completeness assessment of the assembly revealed that 96.1% and 93.7% of the universal single-copy orthologs in the BUSCO embryophyta_odb9 database were present in *R. chiliantha* and *E. roxburghiana*, respectively, indicating that the two genome assemblies were rather complete (Supplementary Table 2).

### Whole-genome duplication

The circular plot and dot plot of the pseudo-chromosomes of *R. chiliantha* and *E. roxburghiana* yielded abundant synteny information

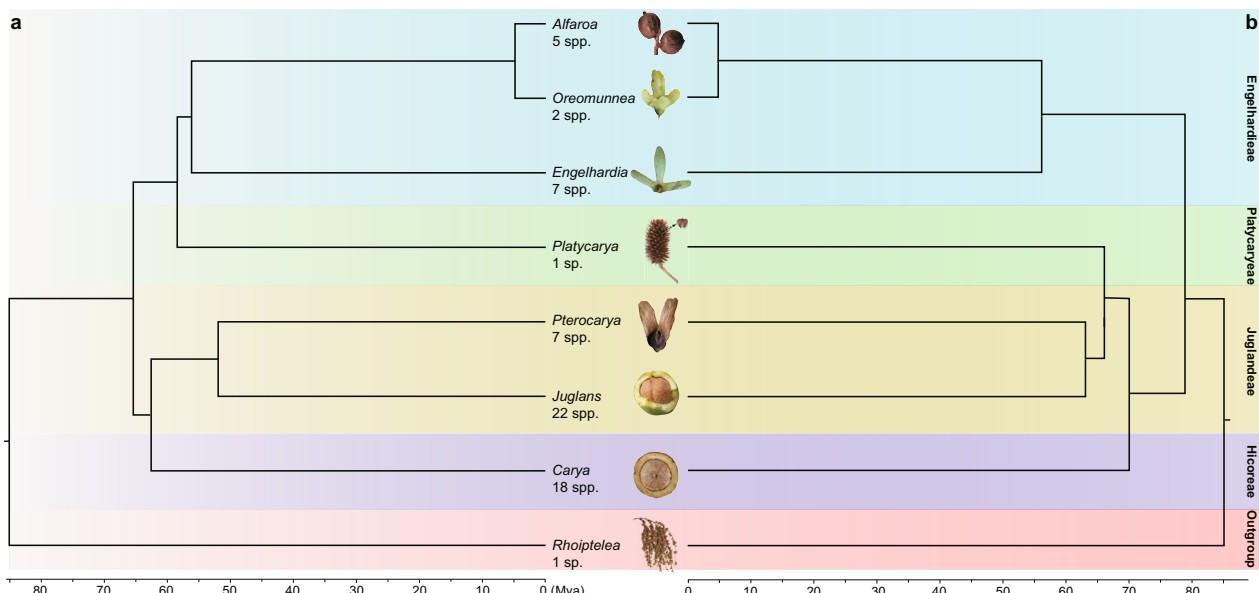

**Fig. 1 | Morphology-based and molecular-based phylogeny of the Juglandaceae. a** Phylogenetic tree modified from Wing and Hickey's[20] cladistic analysis of 30 leaf, fruit, inflorescence, and pollen characters scored for extant and extinct taxa (the latter omitted here), with *Rhoiptelea*, Myricaceae, and Fagaceae as outgroups. For the results of four more recent morphological-cladistic analyses see main text and Supplementary Fig. 17. *Cyclocarya paliurus* (Batal.) Iljinsk. is here treated as *Pterocarya paliurus* following[22,98], and *Alfaropsis roxburghiana* (Wall.) Iljinsk. is treated as *Engelhardia roxburghiana* (compare Supplementary Fig. 16). The fruit photos are not to scale, and the Mya scale is based on the oldest fossil occurrences of the respective genera, but without a formal analysis. **b** The phylogenetic tree on the right is adapted from Mu *et al.*[15] who used nuclear RAD-Seq and whole-chloroplast alignment data.

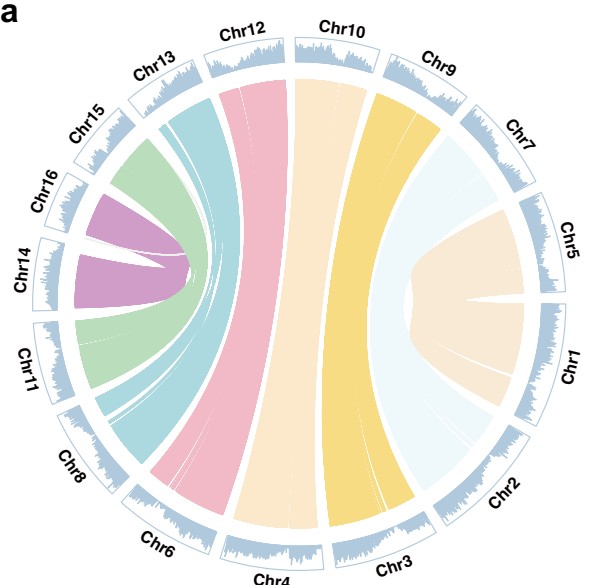

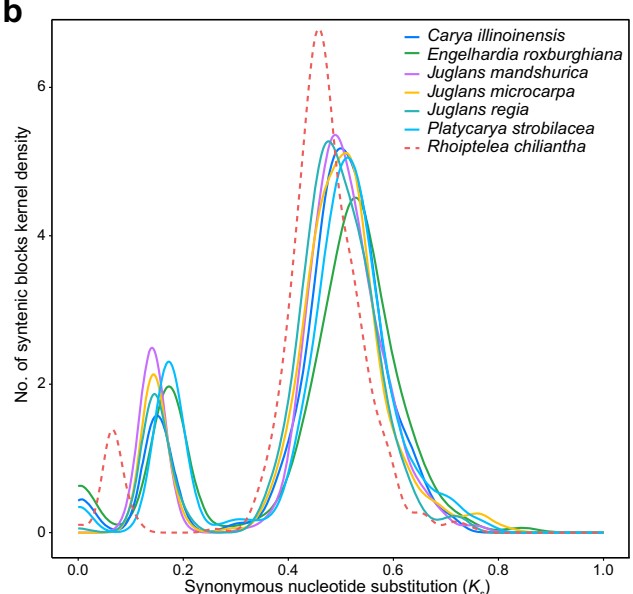

Fig. 2 | **Whole-genome duplication detection. a** A circular plot of *Rhoiptelea chiliantha* homoeologous chromosomes. The reciprocal-best-hits homologous gene pairs were used in MCScanX. Central lines connect collinear blocks across chromosomes, on which gene density is shown. Different colors represent different homoeologous chromosome pairs derived from the most recent shared WGD event. **b** The $K_s$ distribution for the median $K_s$ of each collinear block of seven Juglandaceae species. Source data are provided as a Source Data file.

between homoeologous chromosomes (Fig. 2a and Supplementary Fig. 2, 3). The numbers of collinear blocks (syntenic blocks) were 399 (*Carya illinoinensis*), 403 (*E. roxburghiana*), 324 (*Juglans regia*), 444 (*J. mandshurica*), 338 (*J. microcarpa*), 384 (*Platycarya strobilacea*), and 463 (*R. chiliantha*), respectively. To identify the most recent and more ancient $K_s$ peaks (WGDs) in each species, we plotted the distribution for the median $K_s$ values of the collinear gene pairs contained in each syntenic block within a species. In the $K_s$ distribution, we detected two polyploidization events in each of the genomes of *C. illinoinensis, E. roxburghiana, J. mandshurica, J. microcarpa, J. regia, P. strobilacea* and *R. chiliantha* (Fig. 2b). The first peak of $K_s$ was 0.17 for *R. chiliantha*, whereas it was 0.36–0.48 for the other six species. The second peak of $K_s$ for all species was narrow, pointing to the shared ancient γ-WGT. These results were further supported by the distribution of the four-fold synonymous third-codon transversion rate (4DTv) (Supplementary Fig. 4).

To determine whether *R. chiliantha* shares a recent WGD event with the other Juglandaceae species, we used both genome-synteny analysis[42,43] and the 'multiple gene family' phylogenetic tree method by Pfeil et al. [44], using *J. regia* as a representative of Juglandaceae *sensu stricto*. In the genome-synteny analysis (Fig. 3a), the chromosomes of *R. chiliantha* and *J. regia* formed eight quartets, each consisting of a pair of homoeologous chromosomes from both *R. chiliantha* and *J. regia*. The dot colors clearly distinguish orthologous (yellow: smaller $K_s$) and paralogous (green: larger $K_s$) relationships between interspecific homologous chromosomes, pointing to a shared WGD between them. *Rhoiptelea chiliantha* and the remaining species of Juglandaceae show a similar pattern (Supplementary Fig. 5–9). With Pfeil et al.'s phylogenetic tree method, more than 92% of the gene trees (bootstrap value ≥ 80) supported the topology (((*R. chiliantha, J. regia*), (*R. chiliantha, J. regia*)), *Quercus lobata*), (((*R. chiliantha, J. regia*), *R. chiliantha*), *Q. lobata*) and (((*R. chiliantha, J. regia*), *J. regia*), *Q. lobata*) (Fig. 3b and Supplementary Data 1); the remaining species of Juglandaceae showed a similar pattern (Supplementary Data 1). These two results provide unambiguous evidence that *R. chiliantha* and the remaining Juglandaceae share the same WGD event, which must have been present in their most-recent common ancestor.

## Molecular evolutionary rates following the juglandoid WGD

Although *R. chiliantha* and the other Juglandaceae share the juglandoid WGD, they have markedly different $K_s$ distributions of homoeologous gene pairs. To quantify the discrepancy in the substitution rates of *R. chiliantha* and other Juglandaceae species, we calculated $K_s$ values between a focal species and the common ancestor of its sister clade (for more details see Fig. 3c and its legend). Calculated in this way, $K_s$ now reflects the evolutionary rate of a species since its divergence from the common ancestor (Fig. 3c). A Wilcoxon rank-sum test (Mann-Whitney test) was performed on the $K_s$ distribution of *R. chiliantha* and other species. The results suggested there were significant differences in $K_s$-distribution between *R. chiliantha* and each of other six species ($P < 0.01$; Fig. 3d and Supplementary Fig. 10). We next quantified the rates using Eq. 1, where $t$ is assumed to be 85 MYA based on *Budvaricarpus serialis*, a fossil fruiting structure that is similar to *R. chiliantha*[45] and that is the oldest fossil of Juglandaceae. It comes from the Late Turonian to Santonian, a period spanning 89.8 to 83.6 MYA. The substitution rates were estimated as $1.60 \times 10^{-9}$, $1.95 \times 10^{-9}$, $1.47 \times 10^{-9}$, $1.54 \times 10^{-9}$, $1.51 \times 10^{-9}$, $1.89 \times 10^{-9}$, $0.55 \times 10^{-9}$ per site per year, respectively, for, *C. illinoinensis, E. roxburghiana, J. mandshurica, J. microcarpa, J. regia, P. strobilacea* and *R. chiliantha* (Supplementary Table 3).

## Small effective population size of *Rhoiptelea chiliantha* compared to other Juglandaceae

According to the nearly neutral theory, effective population size ($N_e$) can profoundly influence the rate of molecular evolution[38,46]. Compared with other Juglandaceae species, the effective population size of *R. chiliantha* sharply declined during the time from 5.0 to 0.5 MYA in the PSMC plots (Fig. 4a). Moreover, the $N_e$ of *R. chiliantha* was smaller than that of other species during most of time from 2.0 to 0.1 MYA (Fig. 4a). We also used genome-wide heterozygosity to calculate the long-term average for each species using Eq. 2, where $\mu$ is the mutation rate per year and $g$ is the generation time, assumed to be 30 years (Methods). *Rhoiptelea chiliantha* had the lowest $N_e$ (8,900) among all analyzed species, with the others species' $N_e$ ranging from 10,700 to 25,700 (Supplementary Table 4). We compared the $K_d/K_s$ among

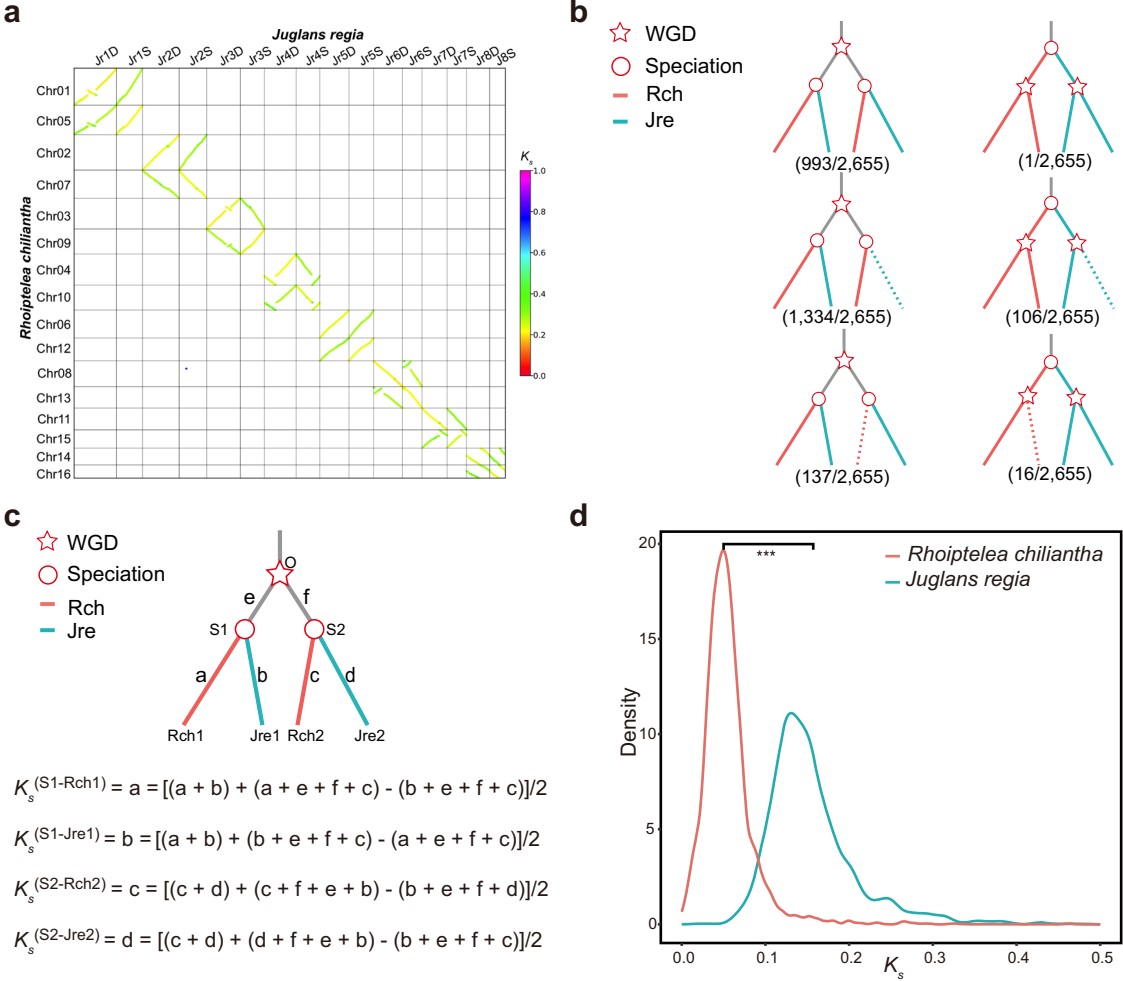

**Fig. 3 | *Rhoiptelea chiliantha* shares the juglandoid whole-genome duplication (WGD) with the rest Juglandaceae species, but exhibits much slower evolution following the polyploidization event. a** Dot plot showing homologous chromosomes of *R. chiliantha* and *J. regia*. Syntenic blocks are plotted with their colors indicating their median $K_s$ values. Each pair of homoeologous chromosomes in one species shows collinear relationships with one homologous pair in another. The dot colors distinguish between orthologous (yellow: smaller $K_s$) and paralogous (green: larger $K_s$) relationships, pointing to the WGD shared by the two species. **b** Phylogenies from the three types of gene families (see main text). The phylogenetic trees at the top, middle, and bottom are from gene families of type 1, 2 and 3, respectively. For the scores shown in parentheses, the denominator is the total number of gene trees with bootstrap values ≥80 (2655 trees) and the numerator represents the number of gene trees displaying the respective topology with

bootstrap values ≥80. The phylogenetic trees on the left (2464 trees) support a shared WGD, those on the right (123 trees) an independent WGD. The red star in (**b**) and (**c**) marks the WGD event, and the red circles indicate speciation events. **c** Relative rate tests used to estimate the evolutionary rate of the different species. Rch1 and Rch2 are the WGD-derived homoeologs of *R. chiliantha*, Jre1 and Jre2 are the WGD-derived homoeologs of *J. regia*; Rch1-Jre1 and Rch2-Jre2 are the orthologous gene pairs due to speciation. $K_s^{(S1-Rch1)}$ is the expected number of substitutions per synonymous site in Rch1 since the speciation event (labeled S1), and other $K_s$ estimates are similarly defined. **d** A comparison of the molecular evolutionary rates for *R. chiliantha* and *J. regia* using the two-tailed two-sample Wilcoxon rank-sum test (Mann–Whitney test). ***$P < 0.001$. Source data are provided as a Source Data file.

lineages and found a significant difference between *R. chiliantha* and the other Juglandaceae species (Fig. 4b and Supplementary Fig. 11). Taken together, these results show that *R. chiliantha* has a small effective population size, which, if anything, would lead to a higher substitution rate, not the slower rate found here.

## More genes related to DNA repair and recombination in *R. chiliantha* compared to other Juglandaceae

*Rhoiptelea chiliantha* has 323 genes identified as being involved in DNA repair and recombination, while the other Juglandaceae have 271 to 302 such genes (Supplementary Data 2 and Supplementary Data 3). These numbers were inferred by assigning the DNA repair and recombination genes from all investigated species to 260 orthogroups (gene families) (Supplementary Data 3). The number of genes of each orthogroup were significantly different between *R. chiliantha* and other species of Juglandaceae (Fig. 4c). We also checked the

asymmetric pattern of DNA repair and recombination genes across orthogroups between *R. chiliantha* and each of the other species. There were 43 orthogroups in which *R. chiliantha* has two gene copies and *J. regia* has at most one gene copy; conversely, there were only seven orthogroups where *J. regia* has two gene copies while *R. chiliantha* has at most one copy. That is to say, there is a strong asymmetry of 43:7 in the comparison of *R. chiliantha* and *J. regia*. Similar numbers of 43:17, 37:14, 34:10, 49:9, and 43:11 were obtained in comparisons of *R. chiliantha* with *E. roxburghiana*, *C. illinoinensis*, *J. mandshurica*, *J. microcarpa* and *P. strobilacea*. There also were 27 orthogroups in which *R. chiliantha* has two copies, while other Juglandaceae species have only one. In 26 of these 27 orthogroups, the two copies in *R. chiliantha* were determined as being derived from the juglandoid WGD, whereas in one orthogroup, the two copies were transposition-derived. All 54 *R. chiliantha* gene copies in the 27 orthogroups were expressed in floral buds, with 45 (83%) of them also

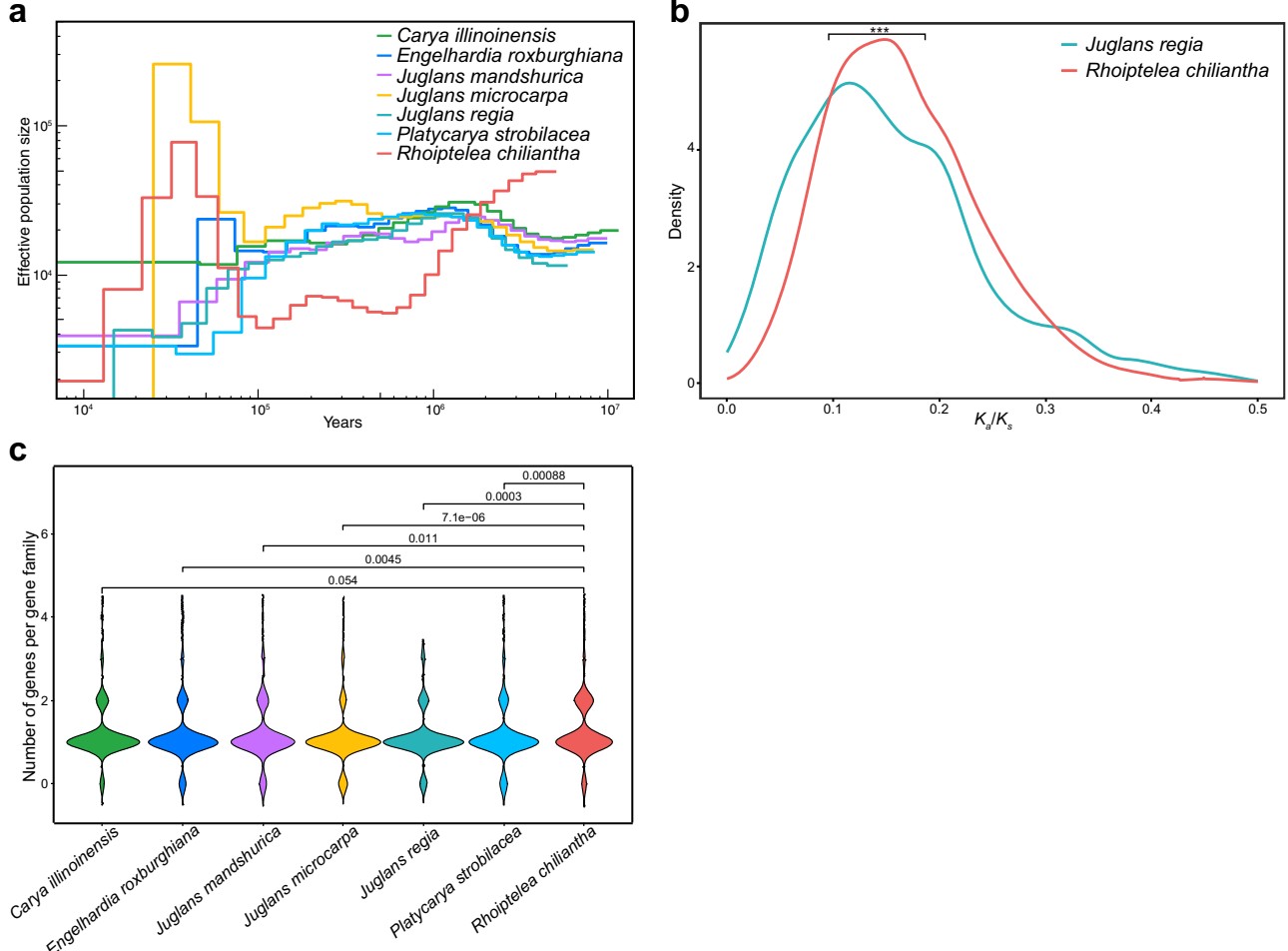

**Fig. 4 | Effective population sizes and the number of genes in gene families associated with DNA repair and recombination. a** Effective population sizes of the seven Juglandaceae species estimated by PSMC. **b** The $K_a/K_s$ distribution for *R. chiliantha* (red) and *J. regia* (blue). $K_a/K_s$ is evaluated as the expected number of substitutions per synonymous site between the focal species and its common ancestor with the other species. The two-tailed two-sample Wilcoxon rank-sum test (Mann–Whitney test) was performed on the $K_a/K_s$ distributions. \*\*\*$P < 0.001$. **c** Violin plots of the number of genes in gene families associated with DNA repair and recombination in the seven Juglandaceae species. The number on the horizontal lines represents the $P$ value of two-tailed two-sample Wilcoxon rank-sum test between *R. chiliantha* and other species. Source data are provided as a Source Data file.

expressed in leaves, suggesting that most, if not all, are truly functional (Supplementary Data 4). By contrast, we found only five orthogroups in which *R. chiliantha* has at most one gene copy but other Juglandaceae species have at least one gene copy (Supplementary Data 3) and six gene copies were expressed in catkin, pistillate flower and leaf of *J. regia* (Supplementary Table 5).

We carried out a KEGG enrichment analysis for the genes that retain two copies in *R. chiliantha* from the juglandoid WGD, but at most only one copy in the other Juglandaceae species. These genes were significantly enriched in transcription-coupled repair (RPB1) and base excision repair (Aprataxin), among various functions (Supplementary Fig. 12). Two recent studies have demonstrated that transcription-coupled repair can play a much larger role in DNA repair than previously thought[47,48].

**Mirror image subgenome-level phylogenies of Juglandaceae**
For the seven Juglandaceae species, we performed subgenome assignments for each pair of homoeologous chromosomes based on the numbers of retained ancestral genes at on the whole chromosome or intraspecific collinear blocks[49–51] (Supplementary Note 1; Supplementary Data 5 and Supplementary Data 6). After subgenome assignment, we used three approaches to infer parental lineages and subgenome relationships in the walnut family, namely microsynteny,

local gene content, and DNA-sequence alignment (numbers of syntenic blocks, matrix lengths, and gene family numbers are given in Supplementary Table 6). Both the microsynteny- and the gene-content-based approach (with the subgenomes assigned by partitioning homoeologous chromosomes) yielded topologies in which the seven recessive subgenomes and *Myrica rubra* formed a clade that was sister to a clade formed by the seven dominant subgenomes (chromosomes with more ancestral genes are here referred to as 'dominant', those with fewer ancestral genes as 'recessive'). Different parameter settings in MCScanX[52], namely $A_5G_{25}$, $A_{10}G_{25}$, or $A_{15}G_{25}$, with 'A' being the minimum number of collinear gene pairs and 'G' the maximum number of intervening genes between adjacent blocks, did not change the topology, consistent with our presupposition that Juglandaceae are of allopolyploid origin (Fig. 5a, b and Supplementary Fig. 13). When the dominant and recessive subgenomes instead were assigned by partitioning intraspecific collinear blocks (allowing us to include one additional species, *Pterocarya stenoptera*, which was assembled at the scaffold level), the $A_5G_{25}$ parameter setting yielded a topology in which the subgenomes formed a clade that was sister to *M. rubra* (Supplementary Fig. 14a), while the $A_{10}G_{25}$ and $A_{15}G_{25}$ settings yielded topologies in which the recessive subgenomes and *Myrica rubra* formed a clade (Supplementary Fig. 14b, c). In all analyses, regardless whether the subgenomes were assigned by partitioning homoeologous

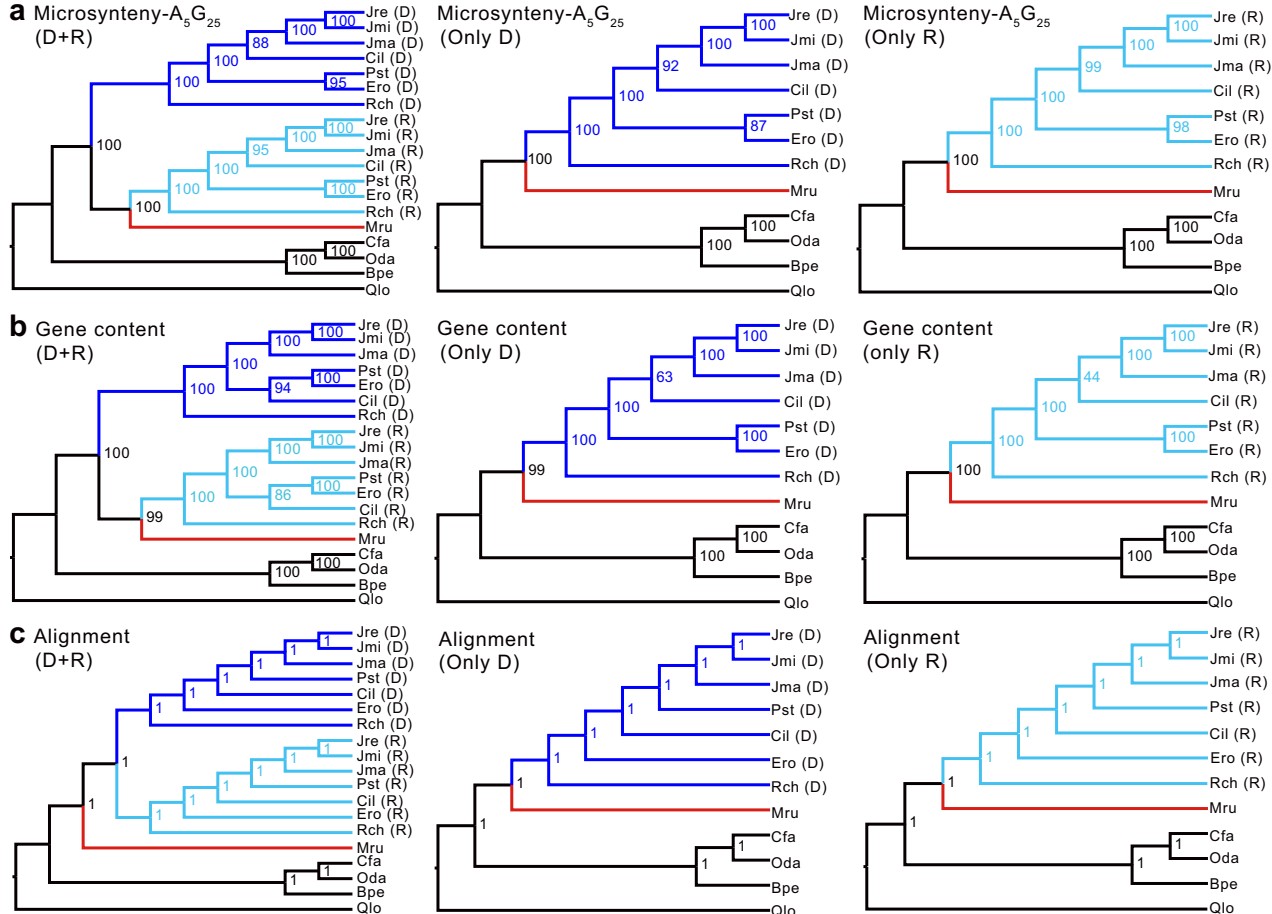

**Fig. 5 | Phylogenetic trees of Juglandaceae obtained from whole-genome microsynteny [Syn-MRL], gene content, and sequence alignments.** The subgenomes assigned by homoeologous chromosomes (dominant and recessive subgenomes of the seven Juglandaceae species and the five outgroups; dominant subgenome (D); recessive subgenome (R)). The Juglandaceae species are *Carya illinoinensis* (Cil), *Engelhardia roxburghiana* (Ero), *Juglans mandshurica* (Jma), *Juglans microcarpa* (Jmi), *Juglans regia* (Jre), *Platycarya strobilacea* (Pst), *Rhoiptelea chiliantha* (Rch) and the outgroups are *Betula pendula* (Bpe), *Carpinus fangiana* (Cfa), *Myrica rubra* (Mru), *Ostryopsis davidiana* (Oda), *Quercus lobata* (Qlo). The panel shows phylogenies inferred by Syn-MRL under $A_5G_{25}$ (A: the minimum number of anchor pairs required to call a collinear block, G: maximum number of intervening genes between two (adjacent) anchor pairs in collinear blocks), (**a**) from microsynteny including both dominant and recessive subgenomes, only dominant subgenomes, or only recessive subgenomes, (**b**) from gene presence/absence or (**c**) DNA-sequence-alignments including both subgenomes, only dominant subgenomes, or only recessive subgenomes. Ultrafast bootstrap (UFBoot) support (%) is shown for each node in (**a**, **b**), and local posterior probability is shown for each internal node in (**c**).

chromosomes or intraspecific collinear blocks, *Platycarya* always grouped with *Engelhardia*, with high bootstrap support (Fig. 5a, b, Supplementary Figs. 13, and Supplementary Fig. 14a–d).

In the trees obtained from the sequence-alignment approach with the subgenomes assigned by partitioning either homoeologous chromosomes or intraspecific collinear blocks, the dominant and recessive subgenomes of Juglandaceae formed a clade that was sister to *M. rubra* (Fig. 5c and Supplementary Fig. 14e) and *Engelhardia* and *Platycarya* were separated from each other (Fig. 5c and Supplementary Fig. 14e).

We also used GRAMPA[53] to identify the most likely placement of the polyploid clade and its parental lineages without a priori subgenome assignment. GRAMPA needs a species tree to start with, which we inferred with ASTRAL-Pro[54]. The optimal multi-labeled tree (MUL-tree) inferred by GRAMPA was consistent with the tree obtained from our subgenome-aware sequence-alignment approach (Supplementary Fig. 15).

## Discussion

The particular placement of the monotypic Asian genus *Platycarya* obtained here from genome-structural data—which remained identical regardless of collinear block sizes or subgenomes used (Fig. 5a, b, Supplementary Fig. 13, and Supplementary Fig. 14a–d)– has not been recovered using DNA alignments, neither in our study (Fig. 5c and Supplementary Fig. 14e) nor over the past 20 years[9,10,14–16,18,19]. This implies that the placement of *Platycarya* in either the genome-structure-based phylogeny or the DNA-alignment-based phylogenies must be wrong. Since the true phylogeny is currently unknown, are there morphological arguments in favor of one or the other placement of *Platycarya*?

Wing and Hickey's[20] analysis of 30 characters scored in fossils and extant species of Juglandaceae yielded a *Platycarya/Engelhardia* group plus its Central American relatives, the genera *Alfaroa* with five Central American species, and *Oreomunnea* with two species in Mexico, neither of them included here. In southeast Asia, *Engelhardia* comprises seven species of which two, *E. roxburgiana* in Vietnam, Laos, Myanmar and southwest China and *E. fenzelii* in southeast China[55], are sometimes recognized as the separate genus *Alfaropsis* (Supplementary Fig. 16 shows a nuclear ITS phylogeny of the entire clade). While Wing and Hickey[20] did not publish their data matrix, their text stresses four to seven leaf architectural traits and pollen and panicle morphology

shared by fossil and living *Platycarya* and *Engelhardia*. In addition, both genera have thin nut walls without lacunae, while the nut walls of *Juglans*, *Carya*, and *Pterocarya* have lacunae[22]. Manchester's[25] intuitive phylogeny of the Juglandaceae showed the same groupings (our Fig. 1a), influenced by Wing and Hickey's study. Four more recent morphological data matrices are available, and we re-analyzed them for this study. Manos and Stone[9] coded 64 characters for 40 living taxa, and their morphological tree left *Platycarya* in a tritomy between the *Engelhardia* clade and the *Juglans* clade (Manos et al. [10]: Fig. 3A). Hermsen and Gandolfo[24] emended and enlarged the matrix of Manos and Stone[9], by coding 64 characters for 28 living and six extinct taxa. Their data yield the (*Platycarya*, *Engelhardia*, *Alfaroa*, *Oreomunnea*) clade, but leave the placement of *Rhoiptelea* unresolved (our Supplementary Fig. 17 shows a Neighbour-Net generated from their data), possibly because they failed to include the oldest fossil of *Rhoiptelea*, an 85 MY old fruiting structure that is the oldest fossil of any Juglandaceae[45]. Larson-Johnson[14] coded 89 characters from 37 living and 27 extinct taxa, and Zhang et al.[19] coded 73 characters for 47 living and 113 extinct taxa. Both matrices fail to unambiguously place *Platycarya*, although fossils of *Platycarya* group with *Engelhardia* in the phylogeny (Fig. 4) of Larson-Johnson[14]. Thus, morphological traits do not contradict the placement of *Platycarya* found here and based on Wing and Hickey[20] may support it (Supplementary Fig. 17).

The DNA-alignment-based trees all placed *Engelhardia* and *Platycarya* far apart (Fig. 5c and Supplementary Fig. 14e). An explanation for this could be post-polyploid gene flow among Juglandaceae (as demonstrated within the genus *Juglans*[26,27]), which would result in discordant phylogenetic signal. A test for gene tree discord that does not require a pre-specified species tree uses gene tree quartets under the multispecies coalescent (MSC) model to calculate quartet count concordance factors (qcCF)[56,57]. We carried out such an analysis for each set of four taxa to evaluate how well expected qcCFs under the MSC model fit the data. With subgenomes assigned via homoeologous chromosomes, we performed MSCquartets analysis on 5,882 gene trees for the seven taxa. Applying the Holm-Bonferroni method to adjust for multiple testing, 12 of 35 tests, about 34.29% (Supplementary Fig. 18 and Supplementary Data 7), rejected the null hypothesis and instead supported significant discord. Of particular note, all quartets, each of which consists of *Engelhardia*, *Platycarya*, *Carya* and one of the *Juglans* species (Supplementary Data 7), produce significant tests, pointing to the possible presence of ancient hybridization between them. Similar results were obtained if the subgenomes were assigned by intraspecific collinear blocks (see Supplementary Fig. 18 and Supplementary Data 8).

As mentioned above, we lack representative genomes of the Central American genera *Alfaroa* (5 species) or *Oreomunnea* (2 species) and a second Asian *Engelhardia*, preferably *E. spicata*, the type species of this genus (Supplementary Fig. 16). Their inclusion could influence the precise attachment of *Platycarya* to the *Engelhardia* clade.

Our results from the $K_s$-analysis, genome synteny (collinearity), and phylogenetics converge to confirm the occurrence of a WGD event in the common ancestor of the walnut family, including *Rhoiptelea* (Fig. 3 and Supplementary Data 1). Our presupposition of an allopolyploidy event at the base of the Juglandaceae family (originally inferred from cytogenetics[7,8]) is supported by the phylogenies that show the recessive subgenome clade as sister to *Myrica* (Fig. 5a–b, Supplementary Figs. 13, and Supplementary Fig. 14b–c), consistent with the result of Xiao et al. [6] who, using sequence similarity as a criterion, found that one subgenome of pecan showed higher average identity with *Myrica rubra*. Reassuringly, a disproportionate loss of duplicated genes between the two subgenomes (i.e., biased fractionation) has been detected in prior studies of *Juglans* and *Carya* genomes[3,5,6] and in our study (Fig. 6 and Supplementary Fig. 19–24). That the recessive subgenome clade is sister to *Myrica* could imply that one parent of Juglandaceae was a species closely related to Myricaceae, while the

other parent was extinct or unsampled. However, the alignment-based tree from both the dominant and recessive subgenomes (Fig. 5c) shows Juglandaceae forming a clade that is sister to *Myrica rubra*, implying that both parents of Juglandaceae were either nested within Myricaceae or had a most recent common ancestor with Myricaceae. Future work should sample an additional genome of Myricaceae, ideally *Canacomyrica monticola*, *Comptonia peregrina*, or *Myrica gale*.

Two $K_s$ peaks corresponding to the juglandoid WGD and the γ-WGT shared by core eudicots were detected in *Rhoiptelea* as well as the other Juglandaceae species (Fig. 2b). The first $K_s$ peak of *R. chiliantha* corresponding to the juglandoid WGD is 2.1 to 2.8 times smaller than that of six other intrafamilial members (Fig. 2b). As Doyle and Egan[58] have pointed out, the $K_s$ peaks of homoeologs in an allopolyploid would overestimate its molecular evolutionary rate because the subgenomes would have diverged at the point of speciation between the two progenitors, rather than at the hybridization (and genome doubling) event itself. So, the $K_s$ for homoeologs derived from a WGD is at best a crude estimate of the descendant lineages' substitution rates (see Thomas et al. [53]: Fig. 1). However, the more direct estimation (Fig. 3c) yields similar results: the rate of *R. chiliantha* was 2.6 to 3.5x slower than that of the other species following the juglandoid WGD.

An extremely slow substitution rate in *R. chiliantha* is supported by additional lines of evidence. The results of the BUSCO assessment (Supplementary Fig. 25) show that 24% of the universal single-copy orthologs in embryophyta_odb10 database were complete and duplicated in *R. chiliantha*, 3–4x more than in the other Juglandaceae species (5.1–6.8%); and most of these duplicate BUSCO groups were derived from the juglandoid WGD (Supplementary Data 9). The percentages of retained orthologous genes of *Rhoiptelea* based on a 100-gene sliding window along each *Q. lobata* chromosome were higher than those in the other six Juglandaceae species for both the dominant subgenome and recessive subgenome (Fig. 6 and Supplementary Fig. 19–24). We also found *R. chiliantha* to retain more WGD gene duplicates (Supplementary Fig. 26 and Supplementary Table 7), indicating that fewer deleterious mutations have accumulated in the *Rhoiptelea* gene duplicates. However, we cannot rule out other hypotheses, such as deletion differences of the genes after WGD[59], calling for additional study.

A species' population size will affect the intensity of selection and drift, which causes different fixation probabilities of nearly neutral mutations in a population. In small populations, weakly deleterious mutations may be retained as effectively neutral alleles, so fewer weakly deleterious mutations are exposed and removed from the gene pool through negative selection, which should lead to high rates of molecular evolution[34–36]. Our results from PSMC, $K_a/K_s$ ratios, and heterozygosity (Fig. 4 and Supplementary Fig. 11), however, indicate that *R. chiliantha* has a smaller $N_e$, no matter in which way it is inferred, making it unlikely that its $N_e$ explains its slower substitution rate. Instead, the higher number of genes in *R. chiliantha* involved in DNA repair and recombination compared to other Juglandaceae (Fig. 4c, a violin plot of the number of genes in gene families associated with DNA repair and recombination in the seven species, and Supplementary Data 3) and the juglandoid WGD genes in *Rhoiptelea* enriched in the transcription-coupled repair (RPB1) and base excision repair (Aprataxin) points to more efficient DNA repair as the principal reason for the reduced substitution rate in *R. chiliantha* following the juglandoid WGD.

In Fig. 2b, the $K_s$ peaks corresponding to the juglandoid WGD range from 0.17 to 0.48, whereas the $K_s$ values corresponding to the γ-WGT range from 1.70 to 2.01. Assuming that the juglandoid WGD occurred at about 85 MYA[45], simple calculations tell us that during the time period after the γ-WGT but before the juglandoid WGD (i.e., from -120 to -85 MYA) the evolutionary rate of Juglandaceae was 7.7 to 22 times higher than that after the juglandoid WGD (i.e., during the time period from 85 MYA up to now). In other words, molecular evolution in

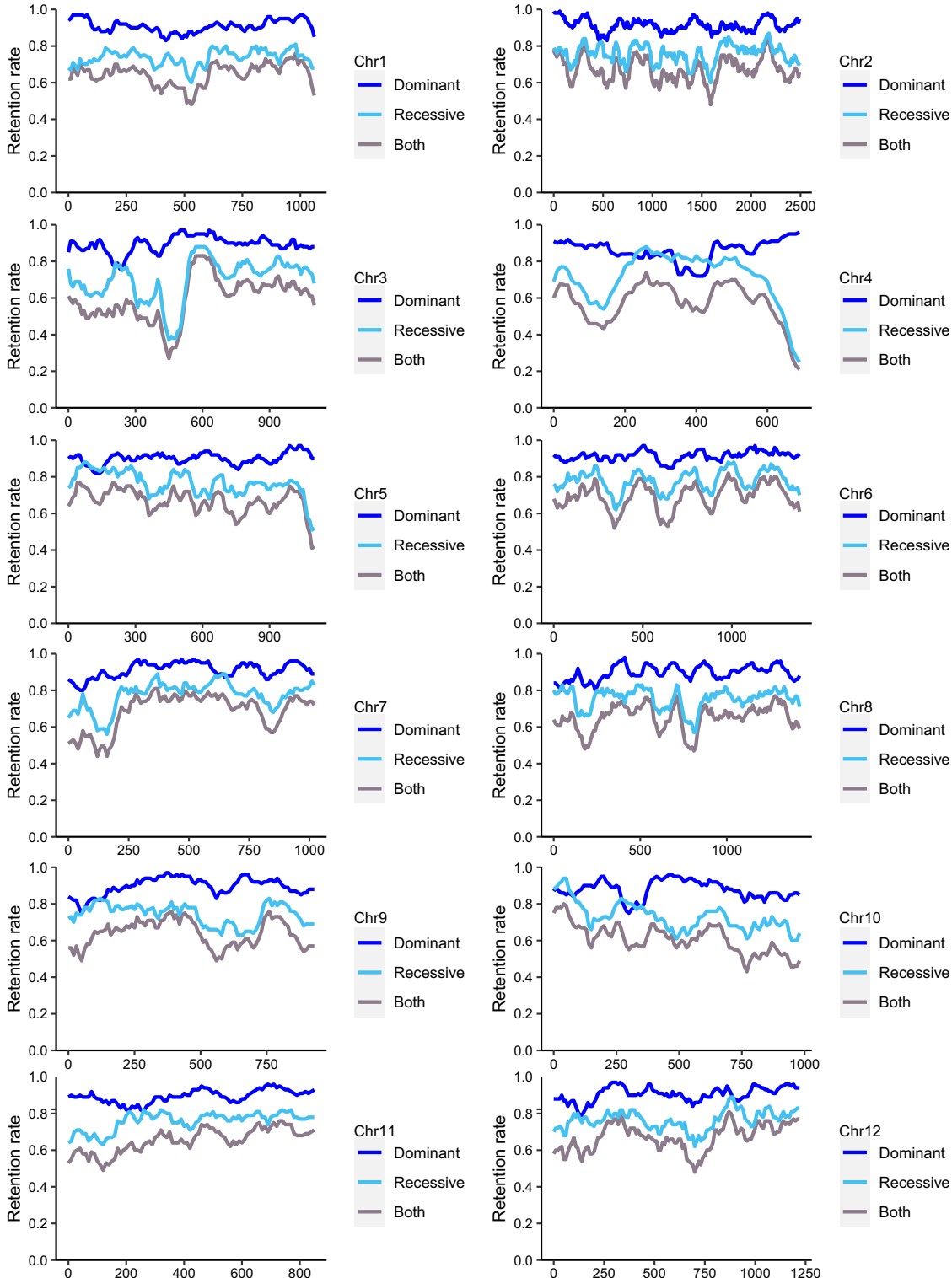

**Fig. 6 | Fractionation pattern on the dominant and recessive subgenome of *Rhoiptelea chiliantha*, using *Quercus lobata* as the target genome.** The X axis indicates gene locations along each *Q. lobata* chromosome, and the Y axis indicates the proportion of orthologous syntenic genes retained (retention rate) in *R. chiliantha* dominant subgenome (blue), recessive subgenome (cyan) and both subgenomes (gray), corresponding to *Q. lobata* chromosomes. The percentage of retained orthologous genes in *R. chiliantha* was calculated based on 100-gene sliding windows along each *Q. lobata* chromosome. Source data are provided as a Source Data file.

Juglandaceae slowed markedly following the juglandoid WGD. Dramatic slowdowns in woody clades of core eudicots were first suggested by Smith and Donoghue[32] and have been quantified in several genomic studies. For instance, the $K_s$ peak values for the salicoid WGD (willow family) occurring at 60–65 MYA[60,61] range from 0.34 to 0.56,

while the $K_s$ peaks for the $\gamma$-WGT of core eudicots to which willows belong were at around 2.5[31], indicating a slowdown of 5.0 times. In the Rosaceae subfamily Pomoideae (containing apples and other fruit trees), the $K_s$ peaks for the Pomoideae WGD at 48–50 MYA[62,63] range from 0.27 to 0.39, compared to the $K_s$ peak for the $\gamma$-WGT at around

2.5[31], indicating a slowdown of 4.5 times. Disregarding such dramatic rate slowdowns results in unreliable estimation of evolutionary events[64], examples being the calibration of the $\gamma$-WGT shared by core eudicots with the time of the juglandoid WGD[6].

In conclusion, the placement of *Platycarya* with *Engelhardia* has implications for the evolution of juglandoid nut walls as well as for the biogeography of the family. Microsynteny and gene-content-based phylogenetic approaches contain so-far undervalued phylogenetic signal. This signal may be especially valuable for studying the phylogenetics of allopolyploid organisms because allopolyploidy involves hybridization and speciation simultaneously. For such lineages, genome-structure permits to distinguish coexisting homeologous chromosomes and to use them separately to infer organismal ancestry. More importantly, any large genome-structural variants capable of altering gene order and/or content probably rarely travel horizontally between species, thus making tree inference based on gene order or content more robust to introgression.

## Methods

### Taxon sampling and genome sequencing, assembling, and downloading

Fresh leaves of *Rhoiptelea chiliantha* (Xichou County, Yunnan Province, China, 23°22′42.73″N, 104°47′17.21″E) and *Engelhardia roxburghiana* (Wangmo County, Guizhou Province, China, 25°15′30.62″N, 105°57′40.7″E) were collected for extracting and sequencing genomic DNA. A permanent voucher of each species has been deposited in the BNU herbarium (*Zhang* BNU20180707-4 and *Cao* BNU20200818-1). *Rhoiptelea chiliantha* and *E. roxburghiana* were sequenced with 44.28 and 103 Gb 150 bp paired-end reads on Illumina HiSeq X Ten sequencing platform, respectively, and GenomeScope 2.0[65] was then used to evaluate their genome sizes with k-mer count histograms constructed by Jellyfish v2.3[66] with a k-mer length of 17. Genome sizes for *Rhoiptelea* and *E. roxburghiana* were ~410 Mbp and ~880 Mbp, respectively (Supplementary Fig. 1).

For *R. chiliantha*, a total of 43 Gb (~105×) PacBio single-molecule long reads and 103 Gb (~108×) Illumina short reads of insert size 350 bp were used for the initial assembly and subsequent correction. In addition, a total of 40.86 Gb (~100×) raw data from Hi-C libraries were used for chromosome-scale genome assembly. For *E. roxburghiana*, a total of 71 Gb (~75×) PacBio single-molecule long reads and 103 Gb (~108×) Illumina short reads of insert size 350 bp were used in the assembly and subsequent correction. A total of 305 Gb (~347×) raw data from Hi-C libraries were used for chromosome-scale genome assembly. The details of each genome assembly are given in Supplementary Note 2.

To represent the major lineages of Juglandaceae (shown in Fig. 1), we downloaded five chromosome-level genomes of *Carya*[67], *Juglans*[3] and *Platycarya*[27] from public data bases. We also downloaded genome of *Myrica rubra* (Myricaceae) and other Fagales that experienced no further polyploidy events beyond the $\gamma$-WGT as the outgroup[68–72] for subsequent analysis. Myricaceae have three genera and 50 species, and are the sister family of Juglandaceae.

### Prediction of repeats and genome annotation

To annotate the genomes of *R. chiliantha* and *E. roxburghiana*, a combination of homology-based inference, ab initio prediction, and transcripts from RNA sequencing (RNA-seq) was used (for details see Supplementary Note 3). The final gene sets were functionally annotated using the Kyoto Encyclopedia of Genes and Genomes (KEGG) Automatic Annotation Server (https://www.genome.jp/tools/kaas/)[73] to perform KEGG Orthology (KO) annotation.

### Detecting whole-genome duplication

We used protein sequence and annotation information from seven species (*C. illinoinensis*, *E. roxburghiana*, *J. mandshurica*, *J. microcarpa*,

*J. regia*, *P. strobilacea* and *R. chiliantha*) to detect and confirm WGD. BLASTP[74] (e < 10$^{-10}$ and top5 matches) was used to search all potential homologous gene pairs of protein sequences within each species and the collinear gene pairs were identified by MCScanX[52] with default settings. We used TBtools[75] to visualize the collinear blocks of 16 chromosomes of *R. chiliantha* and *E. roxburghiana*, respectively. Using information of intra-species collinear gene pairs of the seven species and the inter-species collinear gene pairs between each of the seven species and *Quercus lobata*[69] in DupGen_finder[31], we identified five modes of gene duplication, including whole-genome duplication (WGD), tandem duplication (TD), proximal duplication (PD), transposed duplication (TRD), and dispersed duplication (DSD).

For all collinear gene pairs (anchor genes) of WGD, we estimated synonymous substitutions per synonymous site ($K_s$), nonsynonymous substitutions per nonsynonymous site ($K_a$), and $K_a/K_s$ using Gamma-MYN algorithm[76] in KaKs_Calculator 2.0[77]. The fourfold synonymous third-codon transversion rate (4DTv) between collinear gene pairs was calculated using a Perl script (https://github.com/JinfengChen/Scripts/blob/master/FFgenome/03.evolution/distance_kaks_4dtv/bin/calculate_4DTV_correction.pl). To ensure the independence of $K_s$ and the 4DTv in each collinear block within species, we followed Schnable et al.[49] and estimated the median $K_s$ and 4DTv values for each collinear block. $K_s$ values equal to 0 or greater than 5[31] and 4DTv distances equal to 0 or greater than 1 were excluded. The R package ggplot2[78] was used to plot the distribution of $K_s$ and 4DTv of collinear gene pairs.

### Inference of the WGD event and estimation of the substitution rate for each sampled Juglandaceae species

Based on the $K_s$ distribution of collinear gene pairs, we found that the first peak was at 0.17 in *R. chiliantha*, much smaller than that in the other Juglandaceae where it ranged from 0.36 to 0.48 (Fig. 2b). To account for this discrepancy, we tested two hypotheses: (1) *R. chiliantha* independently experienced a different WGD or (2) *R. chiliantha* shared the same juglandoid WGD but evolved much slower post-WGD. Under the latter hypothesis, the evolutionary rate of each species can be estimated using the procedure developed for the relative rate test[79]. To test if the WGD event is shared or not, we used both intergenomic synteny analysis[42,43] (Fig. 3a) and the 'multiple gene family' approach of Pfeil et al.[44] (Fig. 3b).

For the intergenomic synteny analysis[42,43], we used BLASTP (e < 10$^{-10}$, top5 matches) and MCScanX to determine intergenomic collinear blocks between *R. chiliantha* and the remaining Juglandaceae species. After using the Gamma-MYN algorithm in KaKs_Calculator 2.0 to calculate the $K_s$ of collinear genes, we used WGDI[80] to visualize the median $K_s$ values of collinear blocks between *R. chiliantha* and the remaining Juglandaceae species. If *R. chiliantha* and the other Juglandaceae shared a WGD event, it should precede their divergence from each other; homologous chromosomes between species should show different divergences, and the median $K_s$ values of WGD collinear blocks should be larger than the $K_s$ values of orthologous collinear blocks between *R. chiliantha* and the remaining Juglandaceae species (Fig. 3a). If, however, *R. chiliantha* and the other species experienced independent WGDs, these WGD events would have occurred after their divergence from each other and consequently each chromosome in one species should be equally distant from any homologous chromosome in the other species, producing similar median $K_s$ values for WGD and orthologous collinear blocks.

For the 'multiple gene family' approach[44], we used gene families with at least two gene copies in *R. chiliantha* or the other Juglandaceae species (*C. illinoinensis*, *E. roxburghiana*, *J. mandshurica*, *J. microcarpa*, *J. regia*, *P. strobilacea*) and at least one copy in *Quercus lobata* (see details in Supplementary Note 4 and Supplementary Fig. 27). Within this set of gene families we distinguished three types: (i) families in which both *R. chiliantha* and the other Juglandaceae have two gene copies; (ii) families in which *R. chiliantha* has two copies and the other

species have only one; (iii) families in which *R. chiliantha* has only one copy and the other species have two copies. If *R. chiliantha* and the other species of Juglandaceae (SP) shared the WGD event, the topology of the gene family trees should be either (*Q. lobata*, ((SP, *R. chiliantha*), (SP, *R. chiliantha*))) or (*Q. lobata*, ((SP, *R. chiliantha*), SP)) or (*Q. lobata*, ((SP, *R. chiliantha*), *R. chiliantha*)) (the left part in Fig. 3b). By contrast, if the WGD of *R. chiliantha* and the other species of Juglandaceae (SP) occurred independently, the topology should be either (*Q. lobata*, ((SP, SP), (*R. chiliantha*, *R. chiliantha*))) or (*Q. lobata*, ((SP, SP), *R. chiliantha*)) or (*Q. lobata*, ((*R. chiliantha*, *R. chiliantha*), SP)) (the right part in Fig. 3b).

After establishing that *R. chiliantha* and the other species share the WGD event, we used a relative rate test[79] to estimate the evolutionary rate ($K_s$) for each species (Fig. 3c, Supplementary Fig. 27). The average evolution rate is calculated by

$$r = K_s/t \tag{1}$$

where $r$ is the rate of nucleotide substitution per synonymous site per year, $K_s$ is the expected number of substitutions per synonymous site between the common ancestor and the focal species, and $t$ is the divergence time[64].

### Inferring effective population sizes

To determine whether the seven species of Juglandaceae experienced population contraction/expansion, we used the pairwise sequentially Markovian coalescent (PSMC; v.0.6.5-r67) approach[81] to infer the changes in $N_e$ from the whole-genome information of one diploid individual. Except for *E. roxburghiana* and *R. chiliantha*, the whole-genome sequencing data in this study came from published Juglandaceae genomes used elsewhere (Supplementary Table 1). We used the BWA-mem algorithm[82] of BWA v. 0.7.15 to align trimmed reads from the seven individuals to their respective own reference genomes and used the SENTIEON DNAseq software package v. 201808.08[83] to sort alignment sequence, remove PCR duplicates and realign indels for each individuals. We then used the obtained Binary Alignment Map (BAM) file and SAMTOOLS v.1.2[84] to perform variant calling. SNPs were filtered with the quality adjuster -C setting to 50, the minimal mapping quality to 20. When preparing the input files for PSMC, we set the minimum depth to one third of the mean genome depth and the maximum depth to two times the mean genome depth in vcfutils.pl vcf2fq (https://github.com/lh3/samtools/blob/master/bcftools/vcfutils.pl) pipeline. The substitution rates were set to the values estimated in this study, and the generation time was assumed to be 30 years[85].

We used ANGSD[86] to calculate the heterozygosity of one individual for each species. The BAM file of each individual was as input of ANGSD, with parameters setting as '-C 50 -minQ 20 -minMapQ 30'. The genome-wide heterozygosity was used to estimate the long-term average for each species

$$N_e = \theta/(4 \times \mu \times g) \tag{2}$$

where $\mu$ is the mutation rate per year and $g$ is the generation time.

According to the nearly neutral theory, selection efficiency would be lower in populations with lower effective population size ($N_e$)[46]. Thus, the species with smaller $N_e$ will likely accumulate more nonsynonymous mutations, leading to larger $K_a/K_s$. We compared the distribution of $K_a/K_s$ between *R. chiliantha* and the other Juglandaceae species for those genes that have similar values of $K_s$ in both species.

### Transcriptome mining for genes related to DNA repair and recombination

Four of the major DNA repair pathways: base excision repair (BER), nucleotide excision repair (NER), repair by homologous recombination (HR), and non-homologous end-joining (NHEJ) are highly conserved in eukaryotes[87]. We used KAAS (KEGG Automatic Annotation Server: http://www.genome.jp/kegg/kaas/) to annotate the proteins of the seven Juglandaceae species and then counted the number of genes related to DNA repair and recombination (KEGG Ontology: ko03400). Eleven individuals of two tissues (leaf and flower bud) of *R. chiliantha* were collected for RNA-seq. We also downloaded paired-end RNA-seq data for the leaf, catkin, and pistillate flowers of *J. regia*[2]. After using Trimmomatic v0.32[88] to trim low-quality bases or reads, we used Hisat v.2.1.0[89] to align the trimmed reads to the reference genome and htseq-count v.0.11.2[90] to count the number of reads. The number of fragments per kilobase of transcript per million mapped reads (FPKM) for each gene was calculated with an R script. In this way, we used the RNA-seq data to validate how many genes expressed in *J. regia* and *R. chiliantha* relate to DNA repair and recombination. To infer the function of genes that retain two copies in *R. chiliantha* from the juglandoid WGD, while the other Juglandaceae species have at most one copy, we used R package clusterProfile[91] to perform KEGG enrichment analysis.

### Subgenome assignment and phylogenetic tree inference

To infer the parental species of the juglandoid WGD, we used the subgenomes of seven Juglandaceae species and five representatives of Fagales that have experienced no further WGD beyond the γ-WGT as outgroups, namely *Quercus lobata*[69], *Betula pendula*[70], *Carpinus fangiana*[68], *Ostryopsis davidiana*[71], and *Myrica rubra*[72]. Subgenome assignment was performed for each pair of homoeologous chromosomes (or intraspecific collinear blocks, details in Supplementary Note 1) based on numbers of retained ancestral genes[49–51]. We used reciprocal best hits (RBHs) between *Q. lobata* and each of seven Juglandaceae to obtain a list of retained ancestral genes. We denoted chromosomes with more ancestral genes as 'dominant' and those with fewer ancestral genes as 'recessive'. This approach for assigning subgenomes produces the same result as Zhu et al. [3] who used a slightly different approach based on the number of all genes, and Xiao et al. [6] used the evolutionary distance to assign subgenome.

After subgenome assignment, we applied three methods of phylogenetic inference (alignment-based with ASTRAL-Pro[54], whole-genome microsynteny-based, and local gene content-based) in three datasets to infer the phylogeny of Juglandaceae by including (i) both dominant and recessive subgenomes of all Juglandaceae and the above five outgroups; (ii) only the dominant subgenomes of each Juglandaceae and the outgroups; (iii) only the recessive subgenomes of each Juglandaceae and the outgroups. The protein sequences with the longest transcripts for each gene were selected for further analysis. For the DNA-sequence-alignment method, we used OrthoFinder v.2.4.0[92] to obtain the gene families for each dataset. To avoid multiple sequence alignment errors introduced by too large gene families[93], we kept the size of gene families smaller than 100 genes for downstream analysis. The protein sequences of each gene family were aligned using MAFFT v7.475[94] and then converted into CDS alignment with PAL2NAL v.14[95]. We used IQ-TREE v2.1.2 to construct gene family trees that were then used as input data for ASTRAL-Pro[54] to infer the species tree. Details on the gene families in the three datasets are provided in Supplementary Table 6.

For whole-genome microsynteny-based phylogenetic inference, we used synteny matrix representation with a likelihood (Syn-MRL) pipeline[30] to reconstruct phylogenetic trees from the above three datasets, namely both dominant and recessive subgenomes (dataset 1), only dominant subgenomes (dataset 2), and only recessive subgenomes (dataset 3). Following Zhao et al. [30], BLASTP[74] was used to perform the searches for the all potential inter- and intra-species homologous gene pairs (e < 10⁻¹⁰ and top5 matches). MCScanX[52] was used to detect the synteny blocks of inter- and intra-species with the recommended parameters of A₅G₂₅ (A: the minimum number of

anchor pairs required to call a collinear block, G: maximum number of intervening genes between two (adjacent) anchor pairs in collinear blocks). We also explored the $A_{10}G_{25}$ and $A_{15}G_{25}$ parameter settings, which would have removed the smaller collinear blocks with fewer than 10 or 15 collinear gene pairs from analysis. In the resulting synteny network, nodes are collinear genes in collinear blocks, and edges are used to connect collinear gene pairs. Using the Infomap algorithm v. 1.6.0 (https://github.com/mapequation/infomap), we detected synteny clusters within the map equation framework, setting the two-level partitioning mode with ten trials (--clu -N 10 −2).

For dataset 1 with $A_5G_{25}$ setting, the entire synteny network summarizes information from 67,954 pairwise syntenic blocks, and contains 270,276 nodes and 1,976,821 edges. The entire network was assigned to 19,849 clusters using the Infomap algorithm v. 1.6.0. These clusters were transformed into a binary presence-absence data matrix where rows and columns represent species and clusters, respectively. We removed the cluster that contained only one (sub)genome following Pett et al. [41]. The resulting matrix of 19 rows × 19,202 columns was used to infer a phylogeny with IQ-TREE v2.1.2, under the Mk + R + FO model. Details on the matrices in datasets 2 and 3 with the $A_{10}G_{25}$ and $A_{15}G_{25}$ setting are provided in Supplementary Table 6.

For gene-content-based phylogenetic inference[41], we followed Zhao et al. [30] to assign gene families of the three datasets with OrthoFinder v.2.4.0 and then used the gene presence/absence matrix to construct a phylogenetic tree with IQ-TREE v2.1.2. For each dataset, the gene presence/absence matrices had 19 rows × 17,149 columns, 12 rows × 17,926 columns, and 12 rows × 16,894 columns, respectively.

Since GRAMPA[53] can infer gene duplications and losses in the presence of polyploidy and identify the most likely placement of the polyploid clades and their parental lineages, we also used this method to infer the parental lineages of the juglandoid WGD without performing a priori subgenome assignment. First, we used OrthoFinder v.2.4.0[92] to obtain the gene families for the seven Juglandaceae species and five outgroups and IQ-TREE v2.1.2 to construct gene trees. Then, we used ASTRAL-Pro to infer the species tree from the gene trees. We specified the seven species of Juglandaceae as possible polyploid lineages (H1 nodes) and GRAMPA searched the possible placements of the second parental lineage (H2 nodes). Finally, GRAMPA returned the reconciliation score for each multi-labeled tree (MUL-tree) considered (as well as for the original singly-labeled species tree). The MUL-tree with the minimum reconciliation score was the optimal placement for the H2 node.

### Reporting summary

Further information on research design is available in the Nature Portfolio Reporting Summary linked to this article.

## Data availability

All the sequencing data have been deposited at GenBank under the accession number PRJNA356989. The two assembled genomes have been deposited at GenBank under the accession PRJNA356989 and also available at the website (http://cmb.bnu.edu.cn/juglans). All other raw results related to all figures or tables are available at Figshare[96] (https://doi.org/10.6084/m9.figshare.21901182). Source data are provided with this paper.

## Code availability

All codes for data analysis are provided in the Github[97] [https://github.com/Yamei-Ding/Juglandaceae].

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

## Acknowledgements

The authors thank Mahasin Ali Khan, Sidho-Kanho-Birsha University, for providing images of *Engelhardia* and *Oreomunnea* for Fig. 1 and Guido Grimm, Orléans, for help with morphological analyses and sharing his knowledge of the Fagales fossil record. This work was supported by the National Key R&D Program of China (2017YFA0605104), the National Natural Science Foundation of China (41671040 and 31421063), the "111" Program of Introducing Talents of Discipline to Universities (B13008), and Beijing Advanced Innovation Program for Land Surface Processes.

## Author contributions

W.N.B. and D.Y.Z. conceived of the study. W.P.Z, Y.C., and Y.M.D. performed the sampling and collected the materials. Y.M.D, X.X.P, Y.C., and W.P.Z. performed the analyses. Y.M.D. wrote the first draft of the manuscript. W.N.B., S.S.R., D.Y.Z., Y.M.D., and X.X.P. wrote the revised manuscript. All authors read and approved the final manuscript.

## Competing interests

The authors declare no competing interests.
