## [Peer Review File · Nature Communications]

Juglandaceae phylogenies from genome structure contradict alignment-based inferences, and rate varies with DNA repair genesReviewers' Comments:

Reviewer #1:

Remarks to the Author:

In the manuscript entitled "Genome-structure supports an allotetraploid origin of Juglandaceae from within Myricaceae, and rate variation is proportional to DNA repair genes", Ding and colleagues assembled chromosome-level genomes of two representative species (*Rhoiptelea chiliantha* and *Engelhardia roxburghiana*) in an ancient allopolyploid lineage (walnut family), further reconstruct the phylogenetic history of Juglandaceae and concluded that rate variation for the genome evolution is proportional to DNA repair genes. Here, I have two major concerns for the study.

1. I doubt the conclusion about the correlation between rate variation and the DNA repair genes as I did not see any solid evidence. Is it possible that *Rhoiptelea chiliantha* and the other species could be originated from different allopolyploidization events with different diploid ancestral species and the 'slow rate variation' may not be existed? The author may conduct some analysis to against my opinion.

2. I do not understand the author did not perform the separation for the two subgenomes before the study of "subgenome-level phylogenies". The subgenome-specific short sequences could be used to identify the subgenome (reference: Mitros, T., Session, A.M., James, B.T. et al. Genome biology of the paleotetraploid perennial biomass crop *Miscanthus*. *Nat Commun* 11, 5442 (2020). <https://doi.org/10.1038/s41467-020-18923-6>).

Some minor issues:

1. Abstract, "representative species across an ancient allopolyploid lineage", how many? Two?
2. As a study for genome evolution, the karyotype is essential information for the paper. The author has mentioned the karyotype in line 50, but the whole article has not any karyotype information beside the line 50.
3. Figure 1 is not results from the present study; it is some kind of weird to put it as a single main figure.
4. Fig 2a, 3a, the genomes should be kept in same orientation.
5. Fig. 5c This figure did not present reasonable information for the gene families associated with DNA repair. If your deduction for the rate variation is correct, the information of the 43 orthogroups genes is very importance for study.

Reviewer #2:

Remarks to the Author:

Review of "Genome-structure supports an allotetraploid origin of Juglandaceae from within Myricaceae, and rate variation is proportional to DNA repair genes"

This paper presents some interesting new genomes in the Fagales that will be of importance for agricultural and evolutionary genomics research. The analyses, however, do not in my opinion always well support some of the conclusions made, and more approaches should be considered. I will list some points below in my areas of interest, not necessarily in order of their appearance in the manuscript or their level of importance for consideration of modification/correction.

- Fig. 1; I think it's misleading to show a morphology-based tree as the principal phylogenetic figure. This paper is based on the genomes; I suggest run a coalescence species tree from orthofinder families, or even display the structure-based tree, and then find a way to figure the discordances/concordances - all in one graphic

- Concerning the Ks and 4dtv plots... I am concerned that the kernel densities for the much older (higher Ks [Fig. 2b] or 4dtv) are considerably higher than for the presumed juglandoid WGD. Ancient

gamma peaks are, in my experience, invariably composed of fewer syntenic gene pairs (fractionation preceding continually over time) than more recent events. These results therefore must be thoroughly investigated for the source of this apparent inconsistency. One thing that's needed is self:self dot plots, not just comparative interspecies ones. Then show gamma vs. juglandoid polyploid blocks via their Ks differences. These should also be readily visible in interspecies systemic dot plots too. Why are they not visible in Fig. 3a, for example; were these plots generated from phased subgenome data purged of such blocks? Also, self:self Ks plots can reveal low-Ks haplotigs that can complicate WGD determinations

- Also on Ks. I strongly suggest incorporating *Vitis* as a standard, and *Populus* as a "control" for a species having one WGD. Run Ks and 4dtv the exact same way for these two species. How do the gamma and salicoid polyploidy events behave in direct comparison to those in the newly presented genomes. Is the salicoid peak consistent in Ks with earlier reports, now based on the current investigators' pipeline? Is the gamma peak similarly pronounced more than the later salicoid WGD, comparable to the juglandoid one here? Furthermore, since gamma should be present in all core eudicots, the investigators can try to calibrate all Ks peaks by the mode for syntenic gamma duplicates. But then the investigators make claims of rate shifts post-polyploidy. Also - why not run Ks and 4dtv analyses by subgenome, since they are phased? And why not explicitly study the subgenome split via orthologously systemic gene pair Ks/4dtv?

- Concerning claims for rate slowdowns .. these seem related to WGD dates in the literature, or here for a fossil... were the dates in the literature for Rosaceae and the salicoid WGD themselves derived from rate assumptions? Is there any circular reasoning possible in here?

- Relative rate tests .. these were done on the Ks distributions, correct? What about on the gene families from orthogroup analyses... uniform or pervasive slowdowns here may be more convincing

- On the DNA repair gene family expansions.. I have concern over such functional cherry-picking. Annotations can have biases that turn out to be systemic. If the differences among taxa for syntenic (or tandem) duplicates are truly marked, the functions should show up via GO or other functional enrichments across all annotatable functions - standard fisher's exact tests with correction for multiple tests.

- On the distinct differences between sequence alignment based and synteny/content based phylogenies, if the latter are somehow correct, then the taxon placements there should be reflected in SOME gene trees (the former). I cannot imagine that admixture or ILS could have obscured all gene tree evidence congruent with the structure-based approaches, the latter somehow teasing out something that gene trees never show. In other words, some gene trees really must show synteny-consistent signal. One cannot simply discard the gene tree result in favor of structure-based trees ... one MUST seek, very hard in my opinion, to understand the methodological or biological rationale for the discordance. The structure-based phylogenetic approach is very new! Simulations done and specific cases don't prove its general reliability. It's been mainly ground-truthed on a very taxonomically broad dataset, not so much within a tightly related clade as done here.

- In connection, the authors do not perform any explicit analyses for ancient admixture, e.g., with polyploidy. There is a very nice tool for resolving all-polyploid parentages: GRAMPA (<https://academic.oup.com/sysbio/article/66/6/1007/3610602?login=true>). If the Myricaceae result doesn't pop out here, I wouldn't be so sure about the scenario the authors present. Another approach very likely to be helpful is Whale (<https://academic.oup.com/mbe/article/36/7/1384/5475503?login=true>).. it seems very dangerous to me not to investigate explicitly rather bold interlineage allopolyploid origin hypotheses.

- On the retention of BUSCO duplicates in *Roepetelea* - the authors hypothesis concerning fewer deleterious point mutations largely presumes that fractionation occurs predominantly via

poseudogenization. Analyses across several polyploid systems instead show that deletion differences are the main mechanism, though it may be that initial pseudogenization "targets" given copies for deletional removal. See: <https://www.frontiersin.org/articles/10.3389/fgene.2020.603056/full>

- Concerning subgenome-wise fractionation, and the co-identity of the event observed in *Rhoiptelea* with the others observed, one can directly compare fractionation patterns among the genomes in question, e.g. against *Vitis* as a common denominator, in a setting based on MCScan calculations. One can use FractBias in CoGe (see: <https://academic.oup.com/bioinformatics/article/33/4/552/2617574>)

Reviewer #3:

Remarks to the Author:

In this manuscript, the authors investigated the origin of the walnut family and phylogenetic relationship within it using two newly assembled whole genome data in combination with other published ones. They found an allotetraploid origin of Juglandaceae from within Myricaceae and different phylogenetic position of the genus *Platycarya* from other previous phylogenies. This seems quite surprising to me. However, I am not working on genomics. I cannot judge the solidity of the analysis. I don't have many concerns but some questions. They might be naïve which you can totally ignore.

1. You only used 12 chromosome-level whole genomes of species containing only one species from Myricaceae. I am wondering whether lack of representative species would affect the inference of hybrid event, especially for ancestral ones reported here.
2. In my opinion, phylogenies built based on microsynteny or gene-content characters seem to mostly reflect different patterns of genomic evolution among species after long-term separation, rather than their real phylogenetic relationship. Because the gene-content of one species is affected by many factors such as the soil nutrient etc once the species was established.
3. I do not know how meaningful to compare the effective population sizes through time for species with such different histories. What is the purpose for this analysis? Moreover, it seems *Rhoiptelea* has the most recent origin according to Fig. 5. This is strange because it is the earliest diverged lineage within Juglandaceae.
4. In Fig. 1, the authors said that the scale is based on the oldest fossil occurrences of the respective genera, but not citations in it. As supplementary table of fossil information will be more helpful. You may also map the fossils on the nodes or branches.

Reviewer #4:

Major suggestions

Applying multiple phylogenomic analyses, this study have clarified the phylogenetic position of Engelhardia, the allopolyploid origin of Juglandaceae, and the causes for the slower substitution rates after the juglandoid WGD, which greatly improve the understanding of the evolution of Juglandaceae. My major concerns: 1. What is the major innovation of this study? The authors should highlight their key innovative finding instead of multiple points; 2. The inference of “the entire Juglandaceae apparently arose as a hybrid between two extinct or unsampled progenitors (subgenomes) nested within Myricaceae” maybe rash, the extinct taxa of the ancestor lineage of Myricaceae (only one Mricaceae species was included this analysis) or both Myricaceae and Jugalandaceae maybe the progenitors, it is hard to make a conclusion “nested within Myricaceae; 3. Some writings are not clear, and there are multiple format problems. The manuscript is generally acceptable by NC after major revisions.

Minor suggestions

L29 This sentence sharply turn to talking about Juglandaceae, mentioned in L19, it better to show the family name instead of just mention “the family”.

L52-53 why this sentence?

L54-69 the description of this paragraph is unclear, what is “modern studies”? L54-57 seem show many intergeneric relationships were unresolved, are these conflict relationships are well supported? It is maybe more reasonable to describe just studies using enough molecular data and strong supports, and focused on the unresolved Platycarya. “even when fossil taxa were included (Discussion)”, adding fossil taxa must help to clarify relationships? “also see Manchester²⁷; our Fig. 1”? let reader to read the manuscript of Manchester to know what the paper said? The whole paragraph should be rewrite.

L253 replace “unclear” by “unresolved”.

L273 all italic for “Engelhardia”.

L359 For “Taxon sampling and genome sequencing and assembling”, it is more reasonable to describe the newly sampled and sequenced taxa first, then tell those from previously published ones.

L337 use dash instead of hyphen to show a range, revise across the manuscript.

L360 To represent what of Juglandaceae.

L434 what is “speciation-related collinear blocks”?

L444 the first letter for “Families” should lowercase? Should be consistent as the following.

L462 “each Juglandaceae” what? Species?

L571 some format problem for references

L777 what is “non-cladistic”?

L779 what is the meaning of “mostly unites”? some species are not in this clade or have different dispersed fruits? Clarify!

L780 what is the meaning of “mostly unites”? some species are not in this clade or have different dispersed fruits? Clarify!

L789 a capital letter for “Circular”?

L797 “with”?

REVIEWER COMMENTS

Reviewer #1 (Remarks to the Author):

In the manuscript entitled “Genome-structure supports an allotetraploid origin of Juglandaceae from within Myricaceae, and rate variation is proportional to DNA repair genes”, Ding and colleagues assembled chromosome-level genomes of two representative species (*Rhoiptelea chiliantha* and *Engelhardia roxburghiana*) in an ancient allopolyploid lineage (walnut family), further reconstruct the phylogenetic history of Juglandaceae and concluded that rate variation for the genome evolution is proportional to DNA repair genes. Here, I have two major concerns for the study.

1. I doubt the conclusion about the correlation between rate variation and the DNA repair genes as I did not see any solid evidence. Is it possible that *Rhoiptelea chiliantha* and the other species could be originated from different allopolyploidization events with different diploid ancestral species and the ‘slow rate variation’ may not be existed? The author may conduct some analysis to against my opinion.

Response 1: Because the first peak in the K_s distribution of *Rhoiptelea chiliantha* was at 0.17, much smaller than the peaks in the other Juglandaceae species, which ranged from 0.36 to 0.48 (Fig. 2b), we initially suspected that *R. chiliantha* might have experienced a separate allopolyploidization event from other species of Juglandaceae, just as the reviewer suggests. We then used two methods, genome-synteny analysis (Van de Peer et al., 2009, Trends in Plant Science; Jiao et al., 2014, The Plant Cell) and a phylogenetic tree method based on multiple gene families (Pfeil et al. 2005, Systematic Biology), to determine whether *R. chiliantha* and other Juglandaceae share the same WGD event or not. In the genome-synteny analysis (Fig. 3a), we can see the homologous chromosomes of *R. chiliantha* and *J. regia* (as a representative of the Juglandaceae) forming eight quartets, each quartet consisting of a pair of homoeologous chromosomes from both *R. chiliantha* and *J. regia*. The dot plot clearly shows that orthologous collinear blocks (yellow) between *R. chiliantha* and *J. regia* have smaller K_s whereas the paralogous collinear blocks (green) have larger K_s , indicating that the WGD event precedes the divergence of the species under investigation (Fig. 3a and Supplementary Fig. 5-9). Using the phylogenetic tree method, we found that >92% of the gene trees support the topology (((*R. chiliantha*, *J. regia*), (*R. chiliantha*, *J. regia*)), *Quercus lobata*), (((*R. chiliantha*, *J. regia*), *R. chiliantha*), *Q. lobata*) and (((*R. chiliantha*, *J. regia*), *J. regia*), *Q. lobata*) (Fig. 3b, Supplementary Table 3), providing unambiguous evidence that *R. chiliantha* and the rest Juglandaceae share the same WGD event.

2. I do not understand why the author did not perform the separation for the two subgenomes before the study of “subgenome-level phylogenies”. The subgenome-specific short sequences could be used to identify the subgenome (reference: Mitros, T., Session, A.M., James, B.T. et al. Genome biology of the paleotetraploid perennial

biomass crop *Miscanthus*. Nat Commun 11, 5442 (2020).
<https://doi.org/10.1038/s41467-020-18923-6>).

Response 2: Maybe our description in the main text was too brief. We did perform subgenome assignment for each pair of homoeologous chromosomes, based on the numbers of retained ancestral genes on the whole chromosome or intraspecific collinear blocks (see Supplementary note 1; Supplementary Tables 10 and 11). Our approach for assigning subgenomes follows Schnable et al. (2011, PNAS) and Shi et al. (2020, Molecular Biology and Evolution). This approach produced the same result as the slightly different approach used by Zhu et al. (2019; Horticulture Research) and Xiao et al. (2021; Plant Communications).

Following your advice, we now also tried using the method of Mitros et al. (2020) to identify subgenomes for each species of Juglandaceae. Unfortunately, this method failed to successfully assign subgenomes, probably because the 'juglandoid WGD' was too old (about 85 Mya). Mitros et al. developed their method using the tetraploidy event of *Miscanthus*, which is only ~2.5-6 Mya old.

Some minor issues:

1. Abstract, “representative species across an ancient allopolyploid lineage”, how many? Two?

Response 3: Revised to read, "We assembled two, and downloaded five, chromosome-level genomes of representative species across an ancient allopolyploid lineage".

2. As a study for genome evolution, the karyotype is essential information for the paper. The author has mentioned the karyotype in line 50, but the whole article has not any karyotype information beside the line 50.

Response 4: We have added more karyotype information in the Discussion (see line 305-307).

3. Figure 1 is not results from the present study; it is some kind of weird to put is as a single main figure.

Response 5: We have revised Figure 1 based on your and the second reviewer's suggestion.

4. Fig 2a, 3a, the genomes should be kept in same orientation.

Response 6: Done.

5. Fig. 5c This figure did not present reasonable information for the gene families associated with DNA repair. If your deduction for the rate variation is correct, the

information of the 43 orthogroups genes is very importance for study.

Response 7: Figure 5c suggests that *R. chiliantha* accumulated substitutions more slowly than the other Juglandaceae. To explore the possible causes for this finding, we carried out a KEGG (Kyoto Encyclopedia of Genes and Genomes) enrichment analysis for juglandoid WGD genes of *Rhoiptelea* and also tested alternative explanations, such as population size expansion (see also Responses 13 and 20).

Reviewer #2 (Remarks to the Author):

Review of "Genome-structure supports an allotetraploid origin of Juglandaceae from within Myricaceae, and rate variation is proportional to DNA repair genes"

This paper presents some interesting new genomes in the Fagales that will be of importance for agricultural and evolutionary genomics research. The analyses, however, do not in my opinion always well support some of the conclusions made, and more approaches should be considered. I will list some points below in my areas of interest, not necessarily in order of their appearance in the manuscript or their level of importance for consideration of modification/correction.

Fig. 1; I think it's misleading to show a morphology-based tree as the principal phylogenetic figure. This paper is based on the genomes; I suggest run a coalescence species tree from orthofinder families, or even display the structure-based tree, and then find a way to figure the discordances/concordances - all in one graphic

Response 8: Based on your and the first reviewer's suggestion, we have revised figure 1 to contrast the topology inferred from morphology in 1984 with the topology inferred from DNA-sequence alignments in previous studies and also here. The morphology-based phylogeny from 1984 agrees completely with the phylogenies inferred here from gene content and microsynteny.

Concerning the Ks and 4dTv plots... I am concerned that the kernel densities for the much older (higher Ks [Fig. 2b] or 4dTv) are considerably higher than for the presumed juglandoid WGD. Ancient gamma peaks are, in my experience, invariably composed of fewer syntenic gene pairs (fractionation preceding continually over time) than more recent events. These results therefore must be thoroughly investigated for the source of this apparent inconsistency. One thing that's needed is self:self dot plots, not just comparative interspecies ones. Then show gamma vs. juglandoid polyploid blocks via their Ks differences. These should also be readily visible in interspecies systemic dot plots too. Why are they not visible in Fig. 3a, for example; were these plots generated from phased subgenome data purged of such blocks? Also, self:self Ks plots can reveal low-Ks haplotigs that can complicate WGD determinations.

Response 9: To ensure the independence of Ks and the 4DTv values, we followed Schnable et al.'s (2011; PNAS) method to calculate the median Ks and 4DTv values

for each collinear block and show their distributions. Because of chromosomal rearrangements, the collinear blocks derived from the older eudicot gamma WGT are expected to be more fragmented (i.e. containing fewer syntenic gene pairs, being shorter) and hence relatively more numerous, whereas the collinear blocks derived from the more recent juglandoid WGD are expected to be longer and less numerous. Therefore, the kernel densities for the recent WGD (lower K_s) are considerably lower than the gamma WGT (higher K_s). If all syntenic gene pairs within each collinear block are used to draw the distribution of K_s and 4DTV, the kernel density of the more recent WGD was indeed much higher than that of the gamma WGT (see the figures below). Nonetheless, we followed your advice to add the self:self dot plot in Supplementary Fig. 3, where eight pairs of homoeologous chromosomes in *Rhoiptelea chiliantha* are obvious (also see the figure below). In Figure 3a, we only showed interspecific syntenic blocks with the median K_s values between 0 to 1, so the syntenic blocks with larger K_s (from gamma WGT) are not shown in the dot plot.

The dot plot for *Rhoiptelea chiliantha*.

Also, on Ks. I strongly suggest incorporating *Vitis* as a standard, and *Populus* as a "control" for a species having one WGD. Run Ks and 4dtv the exact same way for these two species. How do the gamma and salicoid polyploidy events behave in direct comparison to those in the newly presented genomes. Is the salicoid peak consistent in Ks with earlier reports, now based on the current investigators' pipeline? Is the gamma peak similarly pronounced more than the later salicoid WGD, comparable to the juglandoid one here? Furthermore, since gamma should be present in all core eudicots, the investigators can try to calibrate all Ks peaks by the mode for syntenic gamma duplicates. But then the investigators make claims of rate shifts post-polyploidy. Also - why not run Ks and 4dtv analyses by subgenome, since they are phased? And why not explicitly study the subgenome split via orthologously systemic gene pair Ks/4dtv?

Response 10: Thanks for your suggestions. We have added *Vitis vinifera* (http://plants.ensembl.org/Vitis_vinifera/Info/Index) and *Populus trichocarpa*

(https://phytozome-next.jgi.doe.gov/info/Ptrichocarpa_v4_1) and calculated the median K_s and the 4DTv for each collinear block (please see the figure below). We detected two polyploidization events in *Populus trichocarpa*, with the first peak of K_s distribution being 0.25 and second peak 2.08, similar to the result of Wei et al. (2020, Horticulture Research).

In Qiao et al. (2019; Genome Biology: Fig. 2a), the K_s peaks corresponding to γ -WGT ranged from 1.91 to 3.64 across 16 core-eudicot taxa, suggesting variable evolution rates post γ -WGT (and among different Juglandaceae species as well; our Fig. 2b). In other words, syntenic gamma duplicates diverged with different rates, necessarily resulting in different values for K_s and 4DTv. As you requested, we also ran K_s and 4DTv analyses between two paleo-subgenomes for each Juglandaceae species, with almost the same result (see the figures below).

Concerning claims for rate slowdowns. these seem related to WGD dates in the literature, or here for a fossil... were the dates in the literature for Rosaceae and the salicoid WGD themselves derived from rate assumptions? Is there any circular reasoning possible in here?

Response 11: Yes, we used the fossil ages to calibrate the timing of the lineage-specific WGD. The fossil record shows that *Populus* and *Salix* diverged by 60 to 65 Ma (Manchester et al., 1986, American Journal of Botany). Thus, the salicoid WGD

event (shared by poplars and willows) is dated as occurring 60 to 65 Ma. As for Pomoideae WGD, we used fossil records of several genera of the Pyreae tribe from the early-middle Eocene flora of western North America, 48-50 Mya (Wolfe and Wehr, 1988, *Aliso: A Journal of Systematic and Floristic Botany*) to time the Pomoideae WGD. In this way of dating WGD, circularity is avoided in our inference.

Relative rate tests .. these were done on the K_s distributions, correct? What about on the gene families from orthogroup analyses... uniform or pervasive slowdowns here may be more convincing

Response 12: Yes, correct, and the relative rate test was based on K_s of the gene families from orthogroup analyses. Indeed, we do observe uniform or pervasive slowdowns in *Rhoiptelea chiliantha*, because its branch length in gene trees is shorter than the other species being compared in this study in almost all cases (~97.0%).

On the DNA repair gene family expansions. I have concern over such functional cherry-picking. Annotations can have biases that turn out to be systemic. If the differences among taxa for syntenic (or tandem) duplicates are truly marked, the functions should show up via GO or other functional enrichments across all annotatable functions - standard fisher's exact tests with correction for multiple tests.

Response 13: Many thanks for your suggestion. We have done KEGG enrichments for the genes that have two copies in *R. chiliantha* from the juglandoid WGD, while other Juglandaceae have at most one copy. These genes were found to be significantly enriched in transcription coupled repair (RPB1) and base excision repair (APTX) (see the figure below).

On the distinct differences between sequence alignment based and synteny/content based phylogenies, if the latter are somehow correct, then the taxon placements there should be reflected in SOME gene trees (the former). I cannot imagine that admixture or ILS could have obscured all gene tree evidence congruent with the structure-based approaches, the latter somehow teasing out something that gene trees never show. In other words, some gene trees really must show synteny-consistent signal. One cannot simply discard the gene tree result in favor of structure-based trees ... one MUST seek, very hard in my opinion, to understand the methodological or biological rationale for the discordance. The structure-based phylogenetic approach is very new! Simulations done and specific cases don't prove its general reliability. It's been mainly ground-truthed on a very taxonomically broad dataset, not so much within a tightly related clade as done here.

Response 14: In our phylogenetic analysis using dominant subgenomes, 1118 single-copy gene trees were used to construct the species tree with ASTRAL-Pro, with 444 (39.7%) having the same topology as the species tree. Only 31 single-copy gene trees

(2.8%) supported the sister-group relationship between *Platycarya* and *Engelhardia* inferred by the microsynteny (and gene-content) approach. We hypothesize that extensive gene flow after the juglandoid WGD may be the principal reason of why sequence alignments and genome-structural information produce conflicting signals. We agree that the microsynteny approach is still in its infancy. Note that in subclades of Brassicaceae that have undergone massive introgression, the synteny-based topology agrees better with the true tree than does the alignment-based topology as illustrated in Zhao et al. (2021; Nature Communications). This example is explained in more detail in response 19 to reviewer #3.

In connection, the authors do not perform any explicit analyses for ancient admixture, e.g., with polyploidy. There is a very nice tool for resolving all-polyploid parentages: GRAMPA (<https://academic.oup.com/sysbio/article/66/6/1007/3610602?login=true>). If the Myricaceae result doesn't pop out here, I wouldn't be so sure about the scenario the authors present. Another approach very likely to be helpful is Whale (<https://academic.oup.com/mbe/article/36/7/1384/5475503?login=true>).. it seems very dangerous to me not to investigate explicitly rather bold interlineage allopolyploid origin hypotheses.

Response 15: We did carry out a GRAMPA analysis and have now added its result to the manuscript (see Supplementary Fig. 14). It yields essentially the same inference about the occurrence and parentage of juglandoid WGD as our present approach (ASTRAL-Pro) based on the subgenome assignment (Fig. 5e).

On the retention of BUSCO duplicates in Rhoiptelea - the authors hypothesis concerning fewer deleterious point mutations largely presumes that fractionation occurs predominantly via pseudogenization. Analyses across several polyploid systems instead show that deletion differences are the main mechanism, though it may be that initial pseudogenization "targets" given copies for deletional removal. See: <https://www.frontiersin.org/articles/10.3389/fgene.2020.603056/full>

Response 16: Thanks for this suggestion. We have added the possible alternative hypothesis of deletion differences in *Discussion* (line 338-339).

Concerning subgenome-wise fractionation, and the co-identity of the event observed in Rhoiptelea with the others observed, one can directly compare fractionation patterns among the genomes in question, e.g. against *Vitis* as a common denominator, in a setting based on MCSScan calculations. One can use FractBias in CoGe (see: <https://academic.oup.com/bioinformatics/article/33/4/552/2617574>)

Response 17: Many thanks for this suggestion. Based on the result of MCSScanX between each of the seven species and *Quercus lobata*, we have displayed the subgenome fractionation pattern for each of the seven Juglandaceae species, with *Quercus lobata* (which, as *Vitis vinifera*, experienced no further polyploidization

event besides γ -WGT) as target genome in the analysis (Fig. 6 and Supplementary Fig. 17-22). Indeed, the percentage of retained orthologous genes for the two paleo-subgenomes of *Rhoiptelea chiliantha* was higher than in other Juglandaceae species, no matter whether *Quercus* or *Vitis* was used as the target genome (see the figure below).

Fractionation pattern of seven species on each reconstructed paleo-subgenome corresponding to *Vitis vinifera* chromosome 1. The X axis represents gene locations along *Vitis vinifera* chromosome 1, and the Y axis represents the proportion of orthologous syntenic genes retained in dominant subgenome (blue), recessive subgenome (cyan) and both subgenomes (gray) of each species, corresponding to *Vitis* chromosome. The percentage of retained orthologous genes in each species was calculated with 100-gene sliding windows along *Vitis* chromosome 1. a) *Rhoiptelea chiliantha*; b) *Engelhardia roxburghiana*; c) *Platycarya strobilacea*; d) *Carya illinoensis*; e) *Juglans regia*; f) *Juglans mandshurica*; g) *Juglans microcarpa*.

Reviewer #3 (Remarks to the Author):

In this manuscript, the authors investigated the origin of the walnut family and phylogenetic relationship within it using two newly assembled whole genome data in combination with other published ones. They found an allotetraploid origin of Juglandaceae from within Myricaceae and different phylogenetic position of the genus *Platycarya* from other previous phylogenies. This seems quite surprising to me. However, I am not working on genomics. I cannot judge the solidity of the analysis. I don't have many concerns but some questions. They might be naïve which you can totally ignore.

1. You only used 12 chromosome-level whole genomes of species containing only one species from Myricaceae. I am wondering whether lack of representative species would affect the inference of hybrid event, especially for ancestral ones reported here.

Response 18: Yes, our sampling of a single species of Myricaceae limits our ability to infer the mode of origin and parental lineages of juglandoid WGD, and we now state this more clearly in the ms. /I inserted a sentence, mentioning the species that Grimm recommended as the best morphological and geographic representatives/ We have also downtoned the title of our ms.

2. In my opinion, phylogenies built based on microsynteny or gene-content characters seem to mostly reflect different patterns of genomic evolution among species after long-term separation, rather than their real phylogenetic relationship. Because the gene-content of one species is affected by many factors such as the soil nutrient etc once the species was established.

Response 19: The microsynteny- and gene-content-based approaches are still in their infancy (cf. reply 14). In the Brassicaceae, alignment-based trees all support genus relationships (A,(B,C)) (the genus names are in Zhao et al., Nature Communications: Supplementary Figs. 15–20), whereas the synteny tree supports the branching order (B,(A,C)), which is the one found in the most recent study of this group of genera (by Forsythe et al. 2020, Genome Biology and Evolution). As reviewer #2 pointed out, however, such specific cases don't prove the general reliability of the new approaches. Nevertheless, there is so far no evidence that microsynteny or gene-content characters only reflect patterns of evolution after long-term separation.

3. I do not know how meaningful to compare the effective population sizes through time for species with such different histories. What is the purpose for this analysis? Moreover, it seems *Rhoiptelea* has the most recent origin according to Fig. 5. This is strange because it is the earliest diverged lineage within Juglandaceae.

Response 20: *Rhoiptelea chiliantha* has a markedly smaller K_s peak value than the other Juglandaceae species which may be caused by its slower evolution rate. According to the nearly neutral theory, effective population size (N_e) can profoundly

affect the rate of molecular evolution (Ohta, 1992, Annual Review of Ecology and Systematics; Lanfear et al., 2014, Trends in Ecology & Evolution). Large N_e would lead to slow evolution rate because of strong selection and weak genetic drift, and vice versa. Therefore, we used PSMC to infer the N_e dynamics for *R. chiliantha*. However, we found that *R. chiliantha* has a smaller N_e , making the hypothesis of larger N_e unlikely to cause slow evolution rate.

We don't understand what the reviewer means by: "Moreover, it seems Rhoiptelea has the most recent origin according to Fig. 5," because PSMC does not provide any inference about the origin of species.

4. In Fig. 1, the authors said that the scale is based on the oldest fossil occurrences of the respective genera, but not citations in it. As supplementary table of fossil information will be more helpful. You may also map the fossils on the nodes or branches.

Response 21: The Mya scale in Fig, 1 is indeed based on the oldest fossil occurrences of the respective genera, but without a formal analysis. We now state this clearly. **The ages of these nodes are not important to our study.**

Reviewer #4: review report attached.

Major suggestions

Applying multiple phylogenomic analyses, this study have clarified the phylogenetic position of Engelhardia, the allopolyploid origin of Juglandaceae, and the causes for the slower substitution rates after the juglandoid WGD, which greatly improve the understanding of the evolution of Juglandaceae. My major concerns: 1. What is the major innovation of this study? The authors should highlight their key innovative finding instead of multiple points.

Response 22: Firstly, using genome-structural information, we recovered the sister group relationship between *Platycarya strobilacea* and the *Engelhardia* clade inferred in a morphological-cladistic analysis (Wing and Hickey, 1984, American Journal of Botany) and accepted in the so-far largest palaeobotanical study of the walnut family (Manchester, 1987, Monograph of Systematic Botany of the Missouri Botanic Garden), but not recovered in four more recent morphological data sets, which we also reanalyzed for this study as explained in our main text and in the legend of Supplementary Fig. 16. All studies that relied on DNA sequence alignments, however, found *Platycarya strobilacea* grouping with the *Juglans* clade. We believe this is because ancient and ongoing hybridization in Juglandaceae influences such large portions of the genome that phylogenetic inference based on DNA sequence alignments will produce species trees that reflect the reticulation history rather than the true bifurcation history (Maddison 1997, Systematic Biology; Mallet et al. 2016, BioEssays). Secondly, we reveal a dramatically slower substitution rate in Rhoiptelea, compared to all other Juglandaceae. We highlight these two key findings in *Introduction and Discussion*.

2. The inference of “the entire Juglandaceae apparently arose as a hybrid between two extinct or unsampled progenitors (subgenomes) nested within Myricaceae” maybe rash, the extinct taxa of the ancestor lineage of Myricaceae (only one Mricaceae species was included this analysis) or both Myricaceae and Jugalandaceae maybe the progenitors, it is hard to make a conclusion “nested within Myricaceae;

Response 23: We have toned down this inference throughout the MS and added a sentence stressing that additional Myricaceae should be sample; we even provide information which Myricaceae would be ideal (see also reply 18).

3. Some writings are not clear, and there are multiple format problems. The manuscript is generally acceptable by NC after major revisions.

Response 24: Thank you. We have corrected the formatting problems.

Minor suggestions

L29 This sentence sharply turn to talking about Juglandaceae, mentioned in L19, it better to show the family name instead of just mention “the family”.

Response 25: Sentence rewritten.

L52-53 why this sentence?

Response 26: Rewritten.

L54-69 the description of this paragraph is unclear, what is “modern studies”?

Response 27: We inserted ‘molecular’.

L54-57 seem show many intergeneric relationships were unresolved, are these conflict relationships are well supported? It is maybe more reasonable to describe just studies using enough molecular data and strong supports, and focused on the unresolved Platycarya. “even when fossil taxa were included (Discussion), adding fossil taxa must help to clarify relationships? “also see Manchester²⁷; our Fig. 1”? let reader to read the manuscript of Manchester to know what the paper said? The whole paragraph should be rewrite.

Response 28: Rewritten.

L253 replace “unclear” by “unresolved”.

Response 29: Done.

L273 all italic for “Engelhardia”.

Response 30: Done.

L359 For “Taxon sampling and genome sequencing and assembling”, it is more reasonable to describe the newly sampled and sequenced taxa first, then tell those from previously published ones.

Response 31: Sequenced changed as suggested.

L337 use dash instead of hyphen to show a range, revise across the manuscript.

Response 32: Done.

L360 To represent what of Juglandaceae.

Response 33: We have rewritten the sentence ‘To represent the major lineages of Juglandaceae (see Fig 1), we downloaded five chromosome-level genomes of *Carya*⁶⁰, *Juglans*³ and *Platycarya*²⁷ from public data bases.’.

L434 what is “speciation-related collinear blocks”?

Response 34: We have changed ‘speciation-related collinear blocks’ into ‘orthologous collinear blocks’.

L444 the first letter for “Families” should lowercase? Should be consistent as the following.

Response 35: Done.

L462 “each Juglandaceae” what? Species?

Response 36: Yes, we mean each Juglandaceae species.

L571 some format problem for references

Response 37: We have checked all the references and corrected the format errors.

L777 what is “non-cladistic”?

Response 38: Wing and Hickey (1984) carried out a formal (computer-based) cladistic analysis. Manchester (1987) was a student in the same cohort as Scott Wing and produced an illustration showing the same topology as found by Wing and Hickey. Our description is correct as it is.

L779 what is the meaning of “mostly unites”? some species are not in this clade or have different dispersed fruits? Clarify!

Response 39: Yes, the genus *Alfaroa* in the *Engelhardia/ Platycarya* clade has animal-dispersed fruits. We now removed the entire sentence.

L780 what is the meaning of “mostly unites”? some species are not in this clade or have different dispersed fruits? Clarify!

Response 40: See response 39.

L789 a capital letter for “Circular”?

Response 41: Done.

L797 “with”?

Response 42: We added the sentence "with the remaining six Juglandaceae species".

Reviewers' Comments:

Reviewer #1:

Remarks to the Author:

The authors have improved the manuscript, but I still have comments

1. Since Fig. 1 was mainly modified from previous studies. In my opinion, it should be presented in the supplementary figures.

2. Ohylogenetic tree is the valid result for testing the differentia of WGD event between *R. chiliantha* and the rest Juglandaceae. Genome-synteny analysis is not sufficient for determining the WGD.

Reviewer #2:

Remarks to the Author:

The authors have done careful checks, analyses and revisions regarding most of my points. The GRAMPA analyses that explicitly support the allopolyploidy events conjectured on other bases are an excellent addition. Adding in *Vitis* and *Populus* to rate calculations was helpful, and showing subgenome fractionation patterns was also a good addition. However, and a big however for me, I remain unimpressed by the authors continued insistence on "trust[ing] the placement of *Platycarya* obtained from genome structure more than that obtained from DNA alignments", as in the quotation below from the revision:

"Our traditional DNA-alignment-based trees all placed *Engelhardia* and *Platycarya* far apart (Fig. 5e, j, o and Supplementary Fig. 13e, j, o). This may be caused by extensive postpolyploid gene flow among Juglandaceae as demonstrated within the genus *Juglans*^{26,27}, which may result in discordant phylogenetic signal. As Zhao et al.³⁰ demonstrated by using the microsynteny approach on 123 fully-sequenced flowering plant genomes, phylogenetic signal from genome structure (gene content, synteny) appears reliable and relatively robust to introgression. Allelic substitutions due to cross-species gene flow can have a large influence on alignment-based inference, but will not impact the gene content and order of the recipient genome, and structural variants, which can profoundly influence gene content and gene order of the introgressed genome, are unlikely to cross species boundary due to strong selection against them⁵⁴. We therefore trust the placement of *Platycarya* obtained from genome structure more than that obtained from DNA alignments."

The authors added the following in their response to my comments:

"Response 14: In our phylogenetic analysis using dominant subgenomes, 1118 singlecopy gene trees were used to construct the species tree with ASTRAL-Pro, with 444 (39.7%) having the same topology as the species tree. Only 31 single-copy gene trees (2.8%) supported the sister-group relationship between *Platycarya* and *Engelhardia* inferred by the microsynteny (and gene-content) approach. We hypothesize that extensive gene flow after the juglandoid WGD may be the principal reason of why sequence alignments and genome-structural information produce conflicting signals. We agree that the microsynteny approach is still in its infancy. Note that in subclades of Brassicaceae that have undergone massive introgression, the synteny-based topology agrees better with the true tree than does the alignment-based topology as illustrated in Zhao et al. (2021; Nature Communications). This example is explained in more detail in response 19 to reviewer #3."

- If the authors wish to invoke cross-lineage/species gene flow as a rationale to prefer synteny-based phylogeny over alignment-based, then they should present explicit analyses for interlineage admixture post- Juglandaceae allopolyploidy. This can be accomplished using Patterson's F3 statistic, which requires no outgroup specification. The fundamentally similar F3-related D statistic has been shown to be robust to deep time divergences (<https://link.springer.com/article/10.1186/s12859-017-2002-4>), and similar F3 analyses for admixture have been done elsewhere among other tree species (see <https://www.nature.com/articles/ng.3862> and <https://www.nature.com/articles/s41588-021-00971->

3). There is an evidential disconnect between invoking massively introgressing Brassicaceae behaving positively in synteny-based phylogenies with the hypothesis that introgression could be confounding alignment-based trees in Juglandaceae. Test this directly, I'd say - with the current data set, not simply citing earlier Juglandaceae studies. SNPs were called for the authors' PSMC work, so F3 analyses (e.g.) should be straightforward. No evidence for admixture would provide no support for their hypothesis of allelic jumbling leading to gene tree incongruence. Note that the point about syntenic information being a conserved phylogenetic characteristic is not in dispute, but the new Zhao et al approach - while interesting and producing broadly consistent results re: angiosperm phylogeny with respect other approaches - remains poorly tested, as I noted in my first review. "Trust" isn't very scientific, even though I fully agree that synteny is, per se, more conservative than ATGC DNA alignments. The Zhao et al. analytical framework, however, is of unclear rationale for why it "works".

Reviewer #3:

Remarks to the Author:

The authors have addressed all of my concerns. I do not have further questions.

Reviewer #4:

Remarks to the Author:

The authors have revised the manuscript following all my suggestions, I have only a few of comments:

A few minor suggestions:

L255, add the reference number after Wing and Hickey.

L264 add the reference number after Manos and Stone

L272 it is better to place Fig. 4 in () after phylogeny.

Beijing and Saint Louis, September 7, 2022

Re: Decision on NCOMMS-22-20452A: "Juglandaceae phylogenies from genome structure contradict alignment-based inferences, and rate variation is proportional to DNA repair genes"

REVIEWER COMMENTS

Reviewer #1 (Remarks to the Author):

The authors have improved the manuscript, but I still have comments. 1. Since Fig. 1 was mainly modified from previous studies. In my opinion, it should be presented in the supplementary figures.

Response 1: Thanks for your suggestion. Although Fig. 1 is mainly modified from previous studies, it is essential to put our work in context, particularly to clarify what is the difference between morphology-based and molecular-based phylogenies of Juglandaceae.

2. Phylogenetic tree is the valid result for testing the differentia of WGD event between *R. chiliantha* and the rest Juglandaceae. Genome-synteny analysis is not sufficient for determining the WGD.

Response 2: Yes, genome-synteny analysis is not sufficient, but nonetheless complementary, for inferring a WGD. We therefore used three methods, namely paralog substitution distributions (plots of the synonymous substitution rate, K_s), phylogenetic trees, and genome-synteny.

Reviewer #2 (Remarks to the Author):

The authors have done careful checks, analyses and revisions regarding most of my points. The GRAMPA analyses that explicitly support the allopolyploidy events conjectured on other bases are an excellent addition. Adding in *Vitis* and *Populus* to rate calculations was helpful, and showing subgenome fractionation patterns was also a good addition. However, and a big however for me, I remain unimpressed by the authors continued insistence on "trust[ing] the placement of *Platycarya* obtained from genome structure more than that obtained from DNA alignments", as in the quotation below from the revision:

"Our traditional DNA-alignment-based trees all placed *Engelhardia* and *Platycarya* far apart (Fig. 5e, j, o and Supplementary Fig. 13e, j, o). This may be caused by extensive postpolyploid gene flow among Juglandaceae as demonstrated within the genus *Juglans*^{26,27}, which may result in discordant phylogenetic signal. As Zhao et al.³⁰ demonstrated by using the microsynteny approach on 123 fully-sequenced flowering plant genomes, phylogenetic signal from genome structure (gene content, synteny) appears reliable and relatively robust to introgression. Allelic substitutions due to cross-species gene flow can have a large influence on alignment-based inference, but

will not impact the gene content and order of the recipient genome, and structural variants, which can profoundly influence gene content and gene order of the introgressed genome, are unlikely to cross species boundary due to strong selection against them⁵⁴. We therefore trust the placement of *Platycarya* obtained from genome structure more than that obtained from DNA alignments."

The authors added the following in their response to my comments:

"Response 14: In our phylogenetic analysis using dominant subgenomes, 1118 single-copy gene trees were used to construct the species tree with ASTRAL-Pro, with 444 (39.7%) having the same topology as the species tree. Only 31 single-copy gene trees (2.8%) supported the sister-group relationship between *Platycarya* and *Engelhardia* inferred by the microsynteny (and gene-content) approach. We hypothesize that extensive gene flow after the juglandoid WGD may be the principal reason of why sequence alignments and genome-structural information produce conflicting signals. We agree that the microsynteny approach is still in its infancy. Note that in subclades of Brassicaceae that have undergone massive introgression, the synteny-based topology agrees better with the true tree than does the alignment-based topology as illustrated in Zhao et al. (2021; Nature Communications). This example is explained in more detail in response 19 to reviewer #3."

- If the authors wish to invoke cross-lineage/species gene flow as a rationale to prefer synteny-based phylogeny over alignment-based, then they should present explicit analyses for interlineage admixture post- Juglandaceae allopolyploidy. This can be accomplished using Patterson's F3 statistic, which requires no outgroup specification. The fundamentally similar F3-related D statistic has been shown to be robust to deep time divergences (<https://link.springer.com/article/10.1186/s12859-017-2002-4>), and similar F3 analyses for admixture have been done elsewhere among other tree species (see <https://www.nature.com/articles/ng.3862> and <https://www.nature.com/articles/s41588-021-00971-3>). There is an evidential disconnect between invoking massively introgressing Brassicaceae behaving positively in synteny-based phylogenies with the hypothesis that introgression could be confounding alignment-based trees in Juglandaceae. Test this directly, I'd say with the current data set, not simply citing earlier Juglandaceae studies. SNPs were called for the authors' PSMC work, so F3 analyses (e.g.) should be straightforward. No evidence for admixture would provide no support for their hypothesis of allelic jumbling leading to gene tree incongruence. Note that the point about syntenic information being a conserved phylogenetic characteristic is not in dispute, but the new Zhao et al approach while interesting and producing broadly consistent results re: angiosperm phylogeny with respect other approaches remains poorly tested, as I noted in my first review. "Trust" isn't very scientific, even though I fully agree that synteny is, per se, more conservative than ATGC DNA alignments. The Zhao et al. analytical framework, however, is of unclear rationale for why it "works".

Response 3: Thank you for this great suggestion. We looked into the various options, and found that F3 analyses need population allele frequencies, which we don't have. Instead, we have now used Patterson's *D*-statistic to detect introgression among the major lineages of Juglandaceae sensu stricto, with *Rhoiptelea chiliantha* as the outgroup. The results have been added to the main text: "To detect whether introgression likely occurred among major lineages of Juglandaceae sensu stricto, we performed Patterson's *D* test^{57,58} on all possible trios of taxa that include *Engelhardia* and *Platycarya* and either *Juglans* (three species) or *Carya* (one species), with *Rhoiptelea chiliantha* as an outgroup (see Supplementary Tables 13 and 14). Given that the species tree is generally unknown for any triplet, we tested introgression on all three possible topologies for each trio of taxa. Only one topology, ((*Juglans regia*, *Platycarya strobilacea*), *Engelhardia roxburghiana*), failed to conform with the conclusion that Patterson's *D* is significantly different from zero (see Supplementary Table 13), implying that ancient introgression must have acted in Juglandaceae sensu stricto." in *Discussion* (lines 277-286).

Reviewer #3 (Remarks to the Author):

The authors have addressed all of my concerns. I do not have further questions.

Reviewer #4 (Remarks to the Author):

The authors have revised the manuscript following all my suggestions, I have only a few of comments:

A few minor suggestions:

L255, add the reference number after Wing and Hickey.

Done.

L264 add the reference number after Manos and Stone

Done.

L272 it is better to place Fig. 4 in () after phylogeny.

Done.

Reviewers' Comments:

Reviewer #1:

Remarks to the Author:

The authors have repensed to all of my concerns.

Reviewer #2:

Remarks to the Author:

3rd review of "Juglandaceae phylogenies from genome structure contradict alignment-based inferences, and rate variation is proportional to DNA repair genes"

Patterson's D-statistic is inappropriate for the purpose raised precisely for the reasons noted by the authors, i.e., "the species tree is generally unknown for any triplet..." and therefore, no proper test using "the three possible topologies for each trio of taxa" ... can be accomplished "with *Rhoiptelea chiliantha* as an outgroup"... given that its use as an outgroup for D-statistics relies on it being a non-admixed taxon itself. Since the authors hypothesize extensive gene flow post- Juglandaceae polyploidy, such an effort would be very unwise using any assumed outgroup. Of course, every possible quartet could be tested using D-statistics, and my group has tried this for other projects; however, interpretation of results can be very difficult, especially if a taxon in the outgroup position of the quartet could be admixed - hence my suggestion to use F3, which requires no outgroup assumption. Unfortunately, D-statistics are often published and interpreted uncritically in the literature as supporting admixture via significantly non-zero Z scores, but the assumptions of the test re: non-admixed outgroup, as noted above, may be rarely met.

Fortunately, however, Patterson's F3 does NOT require population frequencies, although such frequencies can certainly be used as inputs. F3 results are frequently published based on single accession data, most particularly in the "outgroup F3" tests for closest relationship of (e.g., ancient human) accessions via magnitude/significance of positive scores. However, F3 triplets of accessions can be configured in any manner, and some will - through magnitude/significance of negative scores - test admixture, whereas others will test close relationship via identity-by-descent (IBD; for the latter see below under the polar bear paper example).

Here are some examples from population genomics research published in high-impact journals re: the use of F3 to test for admixture among single accessions:

1. Salojärvi J, Smolander OP, Nieminen K, Rajaraman S, Safronov O, Safdari P, Lamminmäki A, Immanen J, Lan T, Tanskanen J, Rastas P. Genome sequencing and population genomic analyses provide insights into the adaptive landscape of silver birch. *Nature genetics*. 2017 Jun;49(6):904-12.

- see, e.g., Fig. 3C

2. Hu G, Feng J, Xiang X, Wang J, Salojärvi J, Liu C, Wu Z, Zhang J, Liang X, Jiang Z, Liu W. Two divergent haplotypes from a highly heterozygous lychee genome suggest independent domestication events for early and late-maturing cultivars. *Nature genetics*. 2022 Jan;54(1):73-83.

- "For cultivated lychees, in the PCA plot, EEMC cultivars clustered with YNW and LMC cultivars with HNW. EMC cultivars, including 'Feizixiao', were distributed intermediately, indicating admixed genetic backgrounds (Fig. 2a,b); this was confirmed by ADMIXTURE analyses (Extended Data Fig. 4) and the formal F3 admixture test"

3. Lan T, Leppälä K, Tomlin C, Talbot SL, Sage GK, Farley SD, Shideler RT, Bachmann L, Wiig Ø, Albert VA, Salojärvi J. Insights into bear evolution from a Pleistocene polar bear genome. *Proceedings*

of the National Academy of Sciences. 2022 Jun 14;119(24):e2200016119.

- see, e.g., Fig. 3B,C,D. Moreover, this paper goes to great lengths demonstrating how F3 operates to evaluate admixture vs. close population co-membership through IBD; see the mathematical proof and discussion here: S15. The f3-statistics - in the supplementary Appendix 01, pp. 13-15.

4. Genomic insights into rapid speciation within the world's largest tree genus [PREPRINT].

YW Low, S Rajaraman, C Tomlin, JA Ahmad, et al - <https://www.researchsquare.com/article/rs-969304/v1> ["Nature Portfolio"], 2021 - now in press in the present journal, Nature Communications

The authors can therefore avoid the issue of outgroup definition entirely by running F3 analysis in all 3-way comparisons of their genomes. If there are significantly negative triplets that cannot be ascribed to close relationship through IBD, then the extensive post-polyploidy admixture hypothesis can hold.

Sadly, few other suitable and explicit approaches exist other than F3. As noted in the "in press" preprint referenced above, mixtures of K components from ADMIXTURE cannot be trusted alone to signify introgression, despite claims by other authors based on K-mixtures observed when running this supervised clustering approach. If one forces K components (ancestral "populations"), one easily obtain K "mixtures" as artifacts of the method. Fro that in-press preprint:

"Next, we used the same SNP data with the ADMIXTURE software⁴⁵ to search for genomic partitioning among the clades and accessions that might be attributable to admixture (introgression) or differential blockwise inheritance through extremely narrow species splits (ILS). ADMIXTURE assumes K ancestral population clusters on the data; it is not decisive regarding mechanisms underlying any K-cluster mixtures within individuals analysed. The approach was developed for population-level data wherein mixed K-clusters are most likely attributable to admixture rather than ILS through lineage splits (e.g., speciation events). However, results at the interspecific level are often interpreted uncritically as actually indicative of cross-lineage admixture (for an example, see Zhang et al., 2020). Indeed, the K components from ADMIXTURE simply represent subsets of inherited SNP variation that could reflect any underlying mixtures, of which ILS can be one mechanistic basis (Supplementary Information, Fig. S9)... It is worth noting that the fold level of cross validation affects the preferred number of components, since the optimum results in a case where for each component there is at least one representative in the test set."

In conclusion, without explicit test results based on the present data that support extensive post-polyploidy introgression, the hypothesis is (in my opinion) quite a stretch. And of course the authors' argument and preference for doubting gene-tree results that differ from their synteny-based phylogenies largely relies on this. My opinion is that the authors still "trust" their synteny-based phylogeny too uncritically (the use of that non-scientific word has not changed in this revision).

Beijing and Saint Louis, September 28, 2022

Re: Decision on NCOMMS-22-20452B: "Juglandaceae phylogenies from genome structure contradict alignment-based inferences, and rate variation is proportional to DNA repair genes"

REVIEWER COMMENTS

Reviewer #1 (Remarks to the Author):

The authors have responded to all of my concerns.

Reviewer #2 (Remarks to the Author): 3rd review of "Juglandaceae phylogenies from genome structure contradict alignment-based inferences, and rate variation is proportional to DNA repair genes"

Patterson's D-statistic is inappropriate for the purpose raised precisely for the reasons noted by the authors, i.e., "the species tree is generally unknown for any triplet..." and therefore, no proper test using "the three possible topologies for each trio of taxa" ... can be accomplished "with *Rhoiptelea chiliantha* as an outgroup" ... given that its use as an outgroup for D-statistics relies on it being a non-admixed taxon itself. Since the authors hypothesize extensive gene flow post- Juglandaceae polyploidy, such an effort would be very unwise using any assumed outgroup. Of course, every possible quartet could be tested using D-statistics, and my group has tried this for other projects; however, interpretation of results can be very difficult, especially if a taxon in the outgroup position of the quartet could be admixed - hence my suggestion to use F3, which requires no outgroup assumption. Unfortunately, D-statistics are often published and interpreted uncritically in the literature as supporting admixture via significantly non-zero Z scores, but the assumptions of the test re: non-admixed outgroup, as noted above, may be rarely met.

Response 1: We agree that a proper application/interpretation of Patterson's *D*-statistic requires a non-admixed outgroup, but please note that our purpose here is to show the existence of post-polyploidization introgression within Juglandaceae. If *Rhoiptelea chiliantha* (the sister to all other living Juglandaceae, which used to be regarded as monotypic family, Rhoipteleaceae) does admix with other Juglandaceae lineages, then it has already met our expectation, i.e., that there is introgression within Juglandaceae. If, instead, it has no admixture with other Juglandaceae, *Rhoiptelea chiliantha* can indeed be used as the outgroup in our use of Patterson's *D*-statistic. Therefore, for our purpose, we needn't worry about the assumption of non-admixing *Rhoiptelea chiliantha*. In other words, a statistically significant Patterson's *D* constitutes sufficient proof of our view that post-polyploid introgression must have acted in Juglandaceae.

Patterson's *D* requires that the rooted species tree is known for a trio of taxa under consideration. For our present purpose of testing if introgression occurs at all, we

adopted a conservative approach by looking at all three possible topologies of each trio. We obtained statistically significant *D*-statistics for all trios of taxa that include *Engelhardia* and *Platycarya* and either *Juglans* (three species) or *Carya* (one species), except for the topology, ((*Juglans regia*, *Platycarya strobilacea*), *Engelhardia roxburghiana*) (see Supplementary Table 13). Therefore, we can safely conclude that introgression has indeed occurred within Juglandaceae.

Reviewer #2 ctd.: Fortunately, however, Patterson's F3 does NOT require population frequencies, although such frequencies can certainly be used as inputs. F3 results are frequently published based on single accession data, most particularly in the "outgroup F3" tests for closest relationship of (e.g., ancient human) accessions via magnitude/significance of positive scores. However, F3 triplets of accessions can be configured in any manner, and some will - through magnitude/significance of negative scores - test admixture, whereas others will test close relationship via identity-by-descent (IBD; for the latter see below under the polar bear paper example).

Here are some examples from population genomics research published in high-impact journals re: the use of F3 to test for admixture among single accessions:

1. Salojärvi J, Smolander OP, Nieminen K, Rajaraman S, Safronov O, Safdari P, Lamminmäki A, Immanen J, Lan T, Tanskanen J, Rastas P. Genome sequencing and population genomic analyses provide insights into the adaptive landscape of silver birch. *Nature genetics*. 2017 Jun;49(6):904-12. - see, e.g., Fig. 3C

2. Hu G, Feng J, Xiang X, Wang J, Salojärvi J, Liu C, Wu Z, Zhang J, Liang X, Jiang Z, Liu W. Two divergent haplotypes from a highly heterozygous lychee genome suggest independent domestication events for early and late-maturing cultivars. *Nature genetics*. 2022 Jan;54(1):73-83.

- "For cultivated lychees, in the PCA plot, EEMC cultivars clustered with YNW and LMC cultivars with HNW. EMC cultivars, including 'Feizixiao', were distributed intermediately, indicating admixed genetic backgrounds (Fig. 2a,b); this was confirmed by ADMIXTURE analyses (Extended Data Fig. 4) and the formal F3 admixture test"

3. Lan T, Leppälä K, Tomlin C, Talbot SL, Sage GK, Farley SD, Shideler RT, Bachmann L, Wiig Ø, Albert VA, Salojärvi J. Insights into bear evolution from a Pleistocene polar bear genome. *Proceedings of the National Academy of Sciences*. 2022 Jun 14;119(24):e2200016119.

- see, e.g., Fig. 3B,C,D. Moreover, this paper goes to great lengths demonstrating how F3 operates to evaluate admixture vs. close population co-membership through IBD; see the mathematical proof and discussion here: S15. The f3-statistics - in the supplementary Appendix 01, pp. 13-15.

4. Genomic insights into rapid speciation within the world's largest tree genus.

YW Low, S Rajaraman, C Tomlin, JA Ahmad, et al - "in press" preprint
<https://www.researchsquare.com/article/rs-969304/v1> [" Nature Portfolio"], 2021 -
now in press in the present journal, Nature Communications

The authors can therefore avoid the issue of outgroup definition entirely by running F3 analysis in all 3-way comparisons of their genomes. If there are significantly negative triplets that cannot be ascribed to close relationship through IBD, then the extensive post-polyploidy admixture hypothesis can hold.

Sadly, few other suitable and explicit approaches exist other than F3. As noted in the "in press" preprint referenced above, mixtures of K components from ADMIXTURE cannot be trusted alone to signify introgression, despite claims by other authors based on K-mixtures observed when running this supervised clustering approach. If one forces K components (ancestral "populations"), one easily obtain K "mixtures" as artifacts of the method. For that in-press preprint: "Next, we used the same SNP data with the ADMIXTURE software⁴⁵ to search for genomic partitioning among the clades and accessions that might be attributable to admixture (introgression) or differential blockwise inheritance through extremely narrow species splits (ILS). ADMIXTURE assumes K ancestral population clusters on the data; it is not decisive regarding mechanisms underlying any K-cluster mixtures within individuals analysed. The approach was developed for population-level data wherein mixed K-clusters are most likely attributable to admixture rather than ILS through lineage splits (e.g., speciation events). However, results at the interspecific level are often interpreted uncritically as actually indicative of cross-lineage admixture (for an example, see Zhang et al., 2020). Indeed, the K components from ADMIXTURE simply represent subsets of inherited SNP variation that could reflect any underlying mixtures, of which ILS can be one mechanistic basis (Supplementary Information, Fig. S9).... It is worth noting that the fold level of cross validation affects the preferred number of components, since the optimum results in a case where for each component there is at least one representative in the test set."

Response 2: Thank you for these suggestions. We have now carried out an admixture f3 test in all triplets of our seven Juglandaceae species and obtained positive values for all f3 statistics (see the following table), providing no support for a scenario of gene flow post-polyploidy. As Patterson et al. (2012, "Ancient admixture in human history, *Genetics* 192: 1065-1093) noted, a history of admixture does not always result in a negative $f_3(C; A, B)$. This can be the case, for example, when target C has experienced a high degree of population-specific drift (due to, in the present context, the deep divergence of the lineages under consideration). We therefore think that the uninformative f3 results are not necessarily in conflict with extensive gene flow after the juglandoid WGD.

Source 1	Source 2	Target	f_3	std. err	Z	SNPs
Juglans regia	Engelhardia roxburghiana	Carya illinoensis	1.368	0.020	69.728	593548
Juglans microcarpa	Engelhardia roxburghiana	Carya illinoensis	1.373	0.020	69.672	606083
Juglans mandshurica	Engelhardia roxburghiana	Carya illinoensis	1.375	0.020	69.852	599845
Platycarya strobilacea	Engelhardia roxburghiana	Carya illinoensis	1.440	0.020	70.698	712575
Rhoiptelea chiliantha	Engelhardia roxburghiana	Carya illinoensis	1.948	0.027	71.496	750643
Juglans regia	Engelhardia roxburghiana	Juglans mandshurica	0.518	0.009	56.427	520658
Juglans regia	Carya illinoensis	Juglans mandshurica	0.526	0.009	57.801	347072
Juglans microcarpa	Engelhardia roxburghiana	Juglans mandshurica	0.542	0.010	56.725	533279
Juglans microcarpa	Carya illinoensis	Juglans mandshurica	0.544	0.009	57.617	359806
Platycarya strobilacea	Carya illinoensis	Juglans mandshurica	0.961	0.015	65.727	499996
Rhoiptelea chiliantha	Carya illinoensis	Juglans mandshurica	0.988	0.015	64.329	572869
Engelhardia roxburghiana	Carya illinoensis	Juglans mandshurica	1.002	0.015	66.172	599845
Carya illinoensis	Engelhardia roxburghiana	Juglans mandshurica	1.002	0.015	66.172	599845
Platycarya strobilacea	Engelhardia roxburghiana	Juglans mandshurica	1.033	0.015	67.134	668511
Rhoiptelea chiliantha	Engelhardia roxburghiana	Juglans mandshurica	1.629	0.024	69.023	708488
Juglans regia	Engelhardia roxburghiana	Juglans microcarpa	0.655	0.011	59.160	525304
Juglans regia	Carya illinoensis	Juglans microcarpa	0.661	0.011	59.761	351374
Juglans regia	Juglans mandshurica	Juglans microcarpa	0.679	0.011	59.629	249210
Platycarya strobilacea	Juglans mandshurica	Juglans microcarpa	0.700	0.012	59.969	430438
Carya illinoensis	Juglans mandshurica	Juglans microcarpa	0.703	0.012	60.274	359806
Juglans mandshurica	Carya illinoensis	Juglans microcarpa	0.703	0.012	60.274	359806
Engelhardia roxburghiana	Juglans mandshurica	Juglans microcarpa	0.705	0.012	60.309	533279
Juglans mandshurica	Engelhardia roxburghiana	Juglans microcarpa	0.705	0.012	60.309	533279

Rhoiptelea chiliantha	Juglans mandshurica	Juglans microcarpa	0.726	0.012	59.876	506525
Platycarya strobilacea	Carya illinoensis	Juglans microcarpa	1.136	0.017	67.007	505686
Engelhardia roxburghiana	Carya illinoensis	Juglans microcarpa	1.185	0.018	67.673	606083
Carya illinoensis	Engelhardia roxburghiana	Juglans microcarpa	1.185	0.018	67.673	606083
Rhoiptelea chiliantha	Carya illinoensis	Juglans microcarpa	1.191	0.018	66.816	580652
Platycarya strobilacea	Engelhardia roxburghiana	Juglans microcarpa	1.214	0.018	68.477	674521
Rhoiptelea chiliantha	Engelhardia roxburghiana	Juglans microcarpa	1.864	0.026	70.859	716198
Juglans mandshurica	Juglans microcarpa	Juglans regia	1.248	0.024	52.355	249210
Juglans microcarpa	Juglans mandshurica	Juglans regia	1.248	0.024	52.355	249210
Carya illinoensis	Juglans microcarpa	Juglans regia	1.279	0.024	52.534	351374
Juglans microcarpa	Carya illinoensis	Juglans regia	1.279	0.024	52.534	351374
Platycarya strobilacea	Juglans microcarpa	Juglans regia	1.286	0.024	52.616	421842
Engelhardia roxburghiana	Juglans microcarpa	Juglans regia	1.289	0.025	52.437	525304
Juglans microcarpa	Engelhardia roxburghiana	Juglans regia	1.289	0.025	52.437	525304
Carya illinoensis	Juglans mandshurica	Juglans regia	1.319	0.025	52.801	347072
Juglans mandshurica	Carya illinoensis	Juglans regia	1.319	0.025	52.801	347072
Platycarya strobilacea	Juglans mandshurica	Juglans regia	1.322	0.025	52.775	417819
Rhoiptelea chiliantha	Juglans microcarpa	Juglans regia	1.330	0.025	52.455	501223
Engelhardia roxburghiana	Juglans mandshurica	Juglans regia	1.333	0.025	52.857	520658
Juglans mandshurica	Engelhardia roxburghiana	Juglans regia	1.333	0.025	52.857	520658
Rhoiptelea chiliantha	Juglans mandshurica	Juglans regia	1.409	0.026	53.211	494946
Platycarya strobilacea	Carya illinoensis	Juglans regia	2.081	0.037	55.911	493157
Engelhardia roxburghiana	Carya illinoensis	Juglans regia	2.165	0.039	56.242	593548
Carya illinoensis	Engelhardia roxburghiana	Juglans regia	2.165	0.039	56.242	593548
Rhoiptelea chiliantha	Carya illinoensis	Juglans regia	2.217	0.039	56.178	569367

Platycarya strobilacea	Engelhardia roxburghiana	Juglans regia	2.221	0.039	56.758	662042
Rhoiptelea chiliantha	Engelhardia roxburghiana	Juglans regia	3.352	0.058	57.594	704489
Rhoiptelea chiliantha	Juglans regia	Platycarya strobilacea	2.521	0.035	72.528	639012
Engelhardia roxburghiana	Juglans regia	Platycarya strobilacea	2.541	0.035	72.955	662042
Juglans regia	Engelhardia roxburghiana	Platycarya strobilacea	2.541	0.035	72.955	662042
Engelhardia roxburghiana	Juglans microcarpa	Platycarya strobilacea	2.542	0.035	72.852	674521
Juglans microcarpa	Engelhardia roxburghiana	Platycarya strobilacea	2.542	0.035	72.852	674521
Rhoiptelea chiliantha	Juglans microcarpa	Platycarya strobilacea	2.545	0.035	72.859	650125
Engelhardia roxburghiana	Juglans mandshurica	Platycarya strobilacea	2.547	0.035	72.955	668511
Juglans mandshurica	Engelhardia roxburghiana	Platycarya strobilacea	2.547	0.035	72.955	668511
Rhoiptelea chiliantha	Juglans mandshurica	Platycarya strobilacea	2.568	0.035	73.035	642447
Engelhardia roxburghiana	Carya illinoensis	Platycarya strobilacea	2.586	0.035	73.028	712575
Carya illinoensis	Engelhardia roxburghiana	Platycarya strobilacea	2.586	0.035	73.028	712575
Rhoiptelea chiliantha	Carya illinoensis	Platycarya strobilacea	2.595	0.035	73.098	687715
Carya illinoensis	Juglans microcarpa	Platycarya strobilacea	2.613	0.036	73.346	505686
Juglans microcarpa	Carya illinoensis	Platycarya strobilacea	2.613	0.036	73.346	505686
Carya illinoensis	Juglans mandshurica	Platycarya strobilacea	2.615	0.036	73.374	499996
Juglans mandshurica	Carya illinoensis	Platycarya strobilacea	2.615	0.036	73.374	499996
Carya illinoensis	Juglans regia	Platycarya strobilacea	2.617	0.036	73.499	493157
Juglans regia	Carya illinoensis	Platycarya strobilacea	2.617	0.036	73.499	493157
Juglans mandshurica	Juglans microcarpa	Platycarya strobilacea	3.009	0.041	74.015	430438
Juglans microcarpa	Juglans mandshurica	Platycarya strobilacea	3.009	0.041	74.015	430438
Juglans mandshurica	Juglans regia	Platycarya strobilacea	3.030	0.041	74.089	417819
Juglans regia	Juglans mandshurica	Platycarya strobilacea	3.030	0.041	74.089	417819
Juglans microcarpa	Juglans regia	Platycarya strobilacea	3.049	0.041	74.220	421842

Juglans regia	Juglans microcarpa	Platycarya strobilacea	3.049	0.041	74.220	421842
Rhoiptelea chiliantha	Engelhardia roxburghiana	Platycarya strobilacea	3.136	0.043	73.476	817612
Engelhardia roxburghiana	Juglans mandshurica	Rhoiptelea chiliantha	157.819	8.220	19.199	708488
Juglans mandshurica	Engelhardia roxburghiana	Rhoiptelea chiliantha	157.819	8.220	19.199	708488
Engelhardia roxburghiana	Carya illinoensis	Rhoiptelea chiliantha	158.340	8.246	19.202	750643
Carya illinoensis	Engelhardia roxburghiana	Rhoiptelea chiliantha	158.340	8.246	19.202	750643
Engelhardia roxburghiana	Juglans microcarpa	Rhoiptelea chiliantha	158.562	8.258	19.201	716198
Juglans microcarpa	Engelhardia roxburghiana	Rhoiptelea chiliantha	158.562	8.258	19.201	716198
Engelhardia roxburghiana	Platycarya strobilacea	Rhoiptelea chiliantha	158.668	8.264	19.199	817612
Platycarya strobilacea	Engelhardia roxburghiana	Rhoiptelea chiliantha	158.668	8.264	19.199	817612
Engelhardia roxburghiana	Juglans regia	Rhoiptelea chiliantha	159.436	8.303	19.202	704489
Juglans regia	Engelhardia roxburghiana	Rhoiptelea chiliantha	159.436	8.303	19.202	704489
Carya illinoensis	Platycarya strobilacea	Rhoiptelea chiliantha	179.807	9.363	19.203	687715
Platycarya strobilacea	Carya illinoensis	Rhoiptelea chiliantha	179.807	9.363	19.203	687715
Juglans mandshurica	Platycarya strobilacea	Rhoiptelea chiliantha	180.834	9.417	19.203	642447
Platycarya strobilacea	Juglans mandshurica	Rhoiptelea chiliantha	180.834	9.417	19.203	642447
Carya illinoensis	Juglans mandshurica	Rhoiptelea chiliantha	181.631	9.457	19.205	572869
Juglans mandshurica	Carya illinoensis	Rhoiptelea chiliantha	181.631	9.457	19.205	572869
Juglans microcarpa	Platycarya strobilacea	Rhoiptelea chiliantha	181.750	9.462	19.208	650125
Platycarya strobilacea	Juglans microcarpa	Rhoiptelea chiliantha	181.750	9.462	19.208	650125
Carya illinoensis	Juglans microcarpa	Rhoiptelea chiliantha	182.459	9.499	19.209	580652
Juglans microcarpa	Carya illinoensis	Rhoiptelea chiliantha	182.459	9.499	19.209	580652
Juglans regia	Platycarya strobilacea	Rhoiptelea chiliantha	182.684	9.511	19.207	639012
Platycarya strobilacea	Juglans regia	Rhoiptelea chiliantha	182.684	9.511	19.207	639012
Carya illinoensis	Juglans regia	Rhoiptelea chiliantha	183.543	9.556	19.208	569367

Juglans regia	Carya illinoensis	Rhoiptelea chiliantha	183.543	9.556	19.208	569367
Juglans mandshurica	Juglans microcarpa	Rhoiptelea chiliantha	198.964	10.356	19.212	506525
Juglans microcarpa	Juglans mandshurica	Rhoiptelea chiliantha	198.964	10.356	19.212	506525
Juglans mandshurica	Juglans regia	Rhoiptelea chiliantha	200.709	10.446	19.214	494946
Juglans regia	Juglans mandshurica	Rhoiptelea chiliantha	200.709	10.446	19.214	494946
Juglans microcarpa	Juglans regia	Rhoiptelea chiliantha	202.375	10.534	19.212	501223
Juglans regia	Juglans microcarpa	Rhoiptelea chiliantha	202.375	10.534	19.212	501223

In conclusion, without explicit test results based on the present data that support extensive post-polyploidy introgression, the hypothesis is (in my opinion) quite a stretch. And of course the authors' argument and preference for doubting gene-tree results that differ from their synteny-based phylogenies largely relies on this. My opinion is that the authors still "trust" their synteny-based phylogeny too uncritically (the use of that non-scientific word has not changed in this revision).

Response 3: We have replaced "trust" with "prefer".

Reviewers' Comments:

Reviewer #2:

Remarks to the Author:

It isn't correct simply that the investigator's significant D statistic result can be casually ascribed to gene flow. Many papers make similar mistakes, however. This is very sloppy, and argument they make below is circular. Firstly, the authors cannot have their cake and eat it too:

"We agree that a proper application/interpretation of Patterson's D-statistic requires a non-admixed outgroup, but please note that our purpose here is to show the existence of post-polyploidization introgression within Juglandaceae."

OK, so they agree that D requires an unadmixed outgroup. Then they add:

"If *Rhoiptelea chiliantha* (the sister to all other living Juglandaceae, which used to be regarded as monotypic family, Rhoipteleaceae) does admix with other Juglandaceae lineages, then it has already met our expectation, i.e., that there is introgression within Juglandaceae. If, instead, it has no admixture with other Juglandaceae, *Rhoiptelea chiliantha* can indeed be used as the outgroup in our use of Patterson's D-statistic. Therefore, for our purpose, we needn't worry about the assumption of non-admixing *Rhoiptelea chiliantha*. In other words, a statistically significant Patterson's D constitutes sufficient proof of our view that post-polyploid introgression must have acted in Juglandaceae. Patterson's D requires that the rooted species tree is known for a trio of taxa under consideration. For our present purpose of testing if introgression occurs at all, we adopted a conservative approach by looking at all three possible topologies of each trio. We obtained statistically significant D-statistics for all trios of taxa that include *Engelhardia* and *Platycarya* and either *Juglans* (three species) or *Carya* (one species), except for the topology, ((*Juglans regia*, *Platycarya strobilacea*), *Engelhardia roxburghiana*) (see Supplementary Table 13). Therefore, we can safely conclude that introgression has indeed occurred within Juglandaceae."

The argument is circular. They admit one can't use the D statistic if the outgroup is admixed → but if *Rhoiptelea* is an admixed outgroup, then their argument is OK (but then the test is invalid) → they get significant D (invalid OR valid test) → so they claim that their test is meaningful, and that introgression occurred in Juglandaceae whether *Rhoiptelea* a valid unadmixedv outgroup or not.

;) To me this is nonsensical, I'm afraid. "We agree that a proper application/interpretation of Patterson's D-statistic requires a non-admixed outgroup"

And then they did F3, actually with INFORMATIVE results - not uninformative ones -

"We have now carried out an admixture f3 test in all triplets of our seven Juglandaceae species and obtained positive values for all f3 statistics (see the following table), providing no support for a scenario of gene flow post-polyploidy."

??

Positive scores provide no evidence FOR admixture. The authors claim the following:

"As Patterson et al. (2012, "Ancient admixture in human history, *Genetics* 192: 1065-1093) noted, a history of admixture does not always result in a negative $f_3(C; A, B)$. This can be the case, for example, when target C has experienced a high degree of population-specific drift (due to, in the present context, the deep divergence of the lineages under consideration). We therefore think that the uninformative f_3 results are not necessarily in conflict with extensive gene flow after the juglandoid

WGD."

What Patterson et al. precisely said in their paper was:

"In Appendix D, we also relax our assumptions about the ascertainment process, showing that F_3 is guaranteed to be positive if C is unadmixed under quite general conditions, for example, polymorphic in the root R and in addition ascertained as polymorphic in any of A, B, C. It is important to recognize, however, that a history of admixture does not always result in a negative $f_3(C; A, B)$ -statistic. If population C has experienced a high degree of population-specific drift (perhaps due to founder events after admixture), it can mask the signal so that $f_3(C; A, B)$ might not be negative."

Patterson et al. describe F_3 further in their appendix,

"If population C is admixed, there is a negative term in the expected value of $f_3(C; A, B)$, which arises because the genetic drift paths $C \rightarrow A$ and $C \rightarrow B$ can take opposite directions through the deepest part of the tree. The observation of a negative value provides unambiguous evidence of population mixture in the history of population C"

Population-specific drift (e.g., "perhaps due to founder events after admixture") is hypothetically possible in the case of some Juglandaceae, but deep divergence is not the population-specific drift referred to here. Regardless, the D-statistic (and by extension, F_4 - another ABBA-BABA formulation .. and then very likely F_3 , which is part of F_4) has been shown to be robust to phylogenetic depth:

Zheng, Y., Janke, A. Gene flow analysis method, the D-statistic, is robust in a wide parameter space. *BMC Bioinformatics* 19, 10 (2018). <https://doi.org/10.1186/s12859-017-2002-4>

Now, I point again to the authors' own results:

"We have now carried out an admixture f_3 test in all triplets of our seven Juglandaceae species and obtained positive values for all f_3 statistics (see the following table), providing no support for a scenario of gene flow post-polyploidy." The actual data are provided as well. No triplet, including all triplets of taxa they use with *Rhoiptelea* as outgroup in D-statistic tests, provide a negative score. Absence of evidence is of course not evidence for absence, but the F_3 results are NOT consistent with the improper use of D, and if anything, conflict with it - no negative scores; no clear evidence for introgression in Juglandaceae history. Again, not ALL admixture scenarios require negative scores, but absence of evidence for introgression except in the case of dubious use of the D statistic (the exact reasons why many investigators dispense with poorly justified D and use F_3 instead!) is not consistent with their hypothesis of Juglandaceae post-polyploid introgression.

The authors can wave their hands about post-polyploid Juglandaceae introgression, but it is in my opinion not demonstrated by them here in a defensible way. And again - they use this unestablished conclusion (I.e., "Therefore, we can safely conclude that introgression has indeed occurred within Juglandaceae.") to justify "trust" in the synteny-based phylogenetic approach over the gene tree approach. Sorry, but I cannot be supportive of this claim. The synteny-based approach can simply be preferred by the authors for any ad hoc rationale, but they cannot admit D might have been improperly applied but then again, if it was it doesn't matter since that would assume *Rhoiptelea* is admixed. Again, IMO, a nonsensical argument.

Please also consider this very recent paper in light of my argument that any potential admixture in *Rhoiptelea* (from any source) would be problematic for D:

Théo Tricou, Eric Tannier, Damien M de Vienne, Ghost Lineages Highly Influence the Interpretation of Introgression Tests, *Systematic Biology*, Volume 71, Issue 5, September 2022, Pages 1147–1158, <https://doi.org/10.1093/sysbio/syac011>

REVIEWER COMMENTS

Reviewer #2 (Remarks to the Author):

It isn't correct simply that the investigator's significant D statistic result can be casually ascribed to gene flow. Many papers make similar mistakes, however. This is very sloppy, and argument they make below is circular. Firstly, the authors cannot have their cake and eat it too: "We agree that a proper application/interpretation of Patterson's D-statistic requires a non-admixed outgroup, but please note that our purpose here is to show the existence of post-polyploidization introgression within Juglandaceae." OK, so they agree that D requires an unadmixed outgroup. Then they add: "If *Rhoiptelea chiliantha* (the sister to all other living Juglandaceae, which used to be regarded as monotypic family, Rhoipteleaceae) does admix with other Juglandaceae lineages, then it has already met our expectation, i.e., that there is introgression within Juglandaceae. If, instead, it has no admixture with other Juglandaceae, *Rhoiptelea chiliantha* can indeed be used as the outgroup in our use of Patterson's D-statistic. Therefore, for our purpose, we needn't worry about the assumption of non-admixing *Rhoiptelea chiliantha*. In other words, a statistically significant Patterson's D constitutes sufficient proof of our view that post-polyploid introgression must have acted in Juglandaceae. Patterson's D requires that the rooted species tree is known for a trio of taxa under consideration. For our present purpose of testing if introgression occurs at all, we adopted a conservative approach by looking at all three possible topologies of each trio. We obtained statistically significant D-statistics for all trios of taxa that include *Engelhardia* and *Platycarya* and either *Juglans* (three species) or *Carya* (one species), except for the topology, ((*Juglans regia*, *Platycarya strobilacea*), *Engelhardia roxburghiana*) (see Supplementary Table 13). Therefore, we can safely conclude that introgression has indeed occurred within Juglandaceae." The argument is circular. They admit one can't use the D statistic if the outgroup is admixed —> but if *Rhoiptelea* is an admixed outgroup, then their argument is OK (but then the test is invalid) —> they get significant D (invalid OR valid test) —> so they claim that their test is meaningful, and that introgression occurred in Juglandaceae whether *Rhoiptelea* a valid unadmixed outgroup or not. To me this is nonessential, I'm afraid. "We agree that a proper application/interpretation of Patterson's D-statistic requires a non-admixed outgroup."

Response 1: We no longer use Patterson's *D*-statistic in this study. Instead, we use a new method, MSCquartets (Rhodes et al. 2021; MSCquartets 1.0: quartet methods for species trees and networks under the multispecies coalescent model in R.

Bioinformatics, 37(12): 1766-1768; Allman et al. 2022. Gene tree discord, simplex plots, and statistical tests under the coalescent. *Systematic Biology*, 71(4):929–942), to test for gene tree discord in the walnut family. MSCquartets does not require a known species-tree and can be a powerful tool for the detection of hybridization. Thus, Børner et al. (2022, Detectability of varied hybridization scenarios using genome-scale hybrid detection methods. *BioRxiv* <https://arxiv.org/abs/2211.00712>) compared summary methods of hybrid detection (*TICR*, *MSCquartets*, *HyDe*, *Patterson's D-Statistic*, *D3*, and *Dp*) and found that MSCquartets outperformed them all under a

range of hybridization scenarios. We have added the results in the *Discussion* (line 277-290) and suppl. Fig. 17.

Supplementary Figure 17. Simplex plots of quartet concordance factors (qcCFs) under the multispecies coalescent (MSC) of ILS and the T3 model (no specific species tree topology hypothesized) using the R package MSCquartets. To minimize gene tree inference error, we perform MSCquartets analyses under different settings for minimum internal quartet branch lengths and after removing such loci that show a signal of recombination. If the branch length was less than 0, 10^{-5} , or 10^{-3} (in substitution unit), quartets were treated as polytomies. PhiPack v 1.1 (Bruen et al., 2006) was used to test for recombination for every gene in the datasets of orthologous genes determined under subgenome assignment at either chromosome level or block level, with a P -value of <0.05 being treated as significant. We used the Holm-Bonferroni method to adjust for multiple testing. Red triangles in the plot represent rejection of the MSC model and indicate gene tree discord mostly caused by introgression; blue circles represent a failure to reject the null hypothesis of MSC model. **(a)-(c)** represent the orthologous genes from the dominant and recessive subgenome of the seven Juglandaceae species whose subgenomes could be assigned by homoeologous chromosomes (including *Carya illinoensis*, *Engelhardia roxburghiana*, *Juglans mandshurica*, *Juglans microcarpa*, *Juglans regia*, *Platycarya strobilacea*, *Rhoiptelea chiliantha*). After removal of recombinant loci, there were 5,882 gene trees for seven taxa and 35 different four-taxon sets in the MSCquartets analysis. **(d)-(f)** represent the orthologous genes from the dominant and recessive subgenomes of the eight Juglandaceae species whose subgenomes could be assigned by intraspecific collinear blocks (including *Carya illinoensis*, *Engelhardia roxburghiana*, *Juglans mandshurica*, *Juglans microcarpa*, *Juglans regia*, *Pterocarya stenoptera*, *Platycarya strobilacea*, *Rhoiptelea chiliantha*). There were 4,679 gene

trees after removing recombinant loci and 70 different four-taxon sets in the MSCquartets analysis.

Reviewer #2: And then they did F3, actually with INFORMATIVE results - not uninformative ones - "We have now carried out an admixture f3 test in all triplets of our seven Juglandaceae species and obtained positive values for all f3 statistics (see the following table), providing no support for a scenario of gene flow post-polyploidy." Positive scores provide no evidence FOR admixture. The authors claim the following: "As Patterson et al. (2012, "Ancient admixture in human history, Genetics 192: 1065-1093) noted, a history of admixture does not always result in a negative f3(C; A, B). This can be the case, for example, when target C has experienced a high degree of population-specific drift (due to, in the present context, the deep divergence of the lineages under consideration). We therefore think that the uninformative f3 results are not necessarily in conflict with extensive gene flow after the juglandoid WGD." What Patterson et al. precisely said in their paper was: "In Appendix D, we also relax our assumptions about the ascertainment process, showing that F3 is guaranteed to be positive if C is unadmixed under quite general conditions, for example, polymorphic in the root R and in addition ascertained as polymorphic in any of A, B, C. It is important to recognize, however, that a history of admixture does not always result in a negative f3(C; A, B)-statistic. If population C has experienced a high degree of population-specific drift (perhaps due to founder events after admixture), it can mask the signal so that f3(C; A, B) might not be negative." Patterson et al. describe F3 further in their appendix, "If population C is admixed, there is a negative term in the expected value of f3(C; A, B), which arises because the genetic drift paths $C \rightarrow A$ and $C \rightarrow B$ can take opposite directions through the deepest part of the tree. The observation of a negative value provides unambiguous evidence of population mixture in the history of population C". Population-specific drift (e.g., "perhaps due to founder events after admixture") is hypothetically possible in the case of some Juglandaceae, but deep divergence is not the population-specific drift referred to here. Regardless, the D-statistic (and by extension, F4 - another ABBA-BABA formulation. and then very likely F3, which is part of F4) has been shown to be robust to phylogenetic depth:

Zheng, Y., Janke, A. Gene flow analysis method, the D-statistic, is robust in a wide parameter space. BMC Bioinformatics 19, 10 (2018). <https://doi.org/10.1186/s12859-017-2002-4>

Response 2: We no longer use Patterson's f3 statistics.

Now, I point again to the authors' own results: "We have now carried out an admixture f3 test in all triplets of our seven Juglandaceae species and obtained positive values for all f3 statistics (see the following table), providing no support for a scenario of gene flow post-polyploidy." The actual data are provided as well. No triplet, including all triplets of taxa they use with *Rhoiptelea* as outgroup in D-statistic tests, provide a negative score. Absence of evidence is of course not evidence for

absence, but the F3 results are NOT consistent with the improper use of D, and if anything, conflict with it - no negative scores; no clear evidence for introgression in Juglandaceae history. Again, not ALL admixture scenarios require negative scores, but absence of evidence for introgression except in the case of dubious use of the D statistic (the exact reasons why many investigators dispense with poorly justified D and use F3 instead!) is not consistent with their hypothesis of Juglandaceae post-polyploid introgression.

The authors can wave their hands about post-polyploid Juglandaceae introgression, but it is in my opinion not demonstrated by them here in a defensible way. And again - they use this unestablished conclusion (I.e., "Therefore, we can safely conclude that introgression has indeed occurred within Juglandaceae.") to justify "trust" in the synteny-based phylogenetic approach over the gene tree approach. Sorry, but I cannot be supportive of this claim. The synteny-based approach can simply be preferred by the authors for any ad hoc rationale, but they cannot admit D might have been improperly applied but then again, if it was it doesn't matter since that would assume *Rhoiptelea* is admixed. Again, IMO, a nonsensical argument. Please also consider this very recent paper in light of my argument that any potential admixture in *Rhoiptelea* (from any source) would be problematic for D:

Théo Tricou, Eric Tannier, Damien M de Vienne, Ghost Lineages Highly Influence the Interpretation of Introgression Tests, *Systematic Biology*, Volume 71, Issue 5, September 2022, Pages 1147–1158, <https://doi.org/10.1093/sysbio/syac011>

Response 3: Thanks for recommending this paper. Following your suggestion, we dropped both previous statistics and now use the above-described quartet method. We detected significant gene tree discord, which most plausibly is due to past hybridization.

Reviewers' Comments:

Reviewer #2:

Remarks to the Author:

The authors now dispense with F statistics based on SNP data and instead focus on gene tree discordance to attempt to make a case for post allopolyploidy hybridization. Ancient admixture is of course one valid explanation for pervasive gene tree discordance. However, it can also point to rampant incomplete lineage sorting (ILS). The authors now use MSCquartets to evaluate gene tree discordance, and find an awful lot of it. Please note that the alternative to hybridization causing species-tree violation is still ILS, and that this may be very difficult to distinguish using MSCquartets on allopolyploid taxa such as those explored here. As such, I would encourage the authors to insert the word "possible" to say instead "... pointing to the POSSIBLE presence of ancient hybridizations between them".. since the results could also point to ILS, even if their preferred explanation is admixture. Note also that the authors have indeed discovered an allopolyploidy (=hybridization) event in their work, and significant discordance from MSCquartets might stem from this hybridization when quartet counts from within their larger gene trees are explored by the method. Thus, if it were me, I would couch my language on the preference for post-polyploid introgression accounting for the discordance - and my insertion of the word POSSIBLE above would do that I think... indeed, even the MSCquartets authors couch their own use of the word hybridization in their paper describing the method (note the word PERHAPS, below): "With exhaustive alternative hypotheses, rejection of the null indicates in the first case that the MSC on the specific species tree topology is rejected, and in the second that the MSC on any tree is rejected (PERHAPS due to hybridization).."

Reviewer #4:

Remarks to the Author:

I agree with reviewer 2 that the rejection of the null hypothesis in MSCQuartets does not automatically imply hybridization as there are many sources of gene tree discordance that are not properly explained by the tree-based coalescent model. That is, the null hypothesis in MSCQuartets is that a coalescent model on a tree (that accounts solely for ILS as source of gene tree discordance) is enough to explain the gene tree discordance observed in the data. Rejecting this claim means that ILS is not enough to explain the observed gene tree discordance, yet we are unable to conclude with certainty that the alternative has to be hybridization. Combined with other analyses, the authors might have strong suspicion that hybridization is the cause for discordance, but this conclusion cannot be reached from MSCQuartets alone.

I echo the reviewer with the addition of the word "possible".

Minor comment:

- Line 277: "that does not require a species tree". MSCQuartets uses intrinsically a species tree to obtain the expected CFs under the coalescent model. Maybe I would change this sentence to "that does not require a pre-specified species tree"

REVIEWERS' COMMENTS

Reviewer #2 (Remarks to the Author):

The authors now dispense with F statistics based on SNP data and instead focus on gene tree discordance to attempt to make a case for post allopolyploidy hybridization. Ancient admixture is of course one valid explanation for pervasive gene tree discordance. However, it can also point to rampant incomplete lineage sorting (ILS). The authors now use MSCquartets to evaluate gene tree discordance, and find an awful lot of it. Please note that the alternative to hybridization causing species-tree violation is still ILS, and that this may be very difficult to distinguish using MSCquartets on allopolyploid taxa such as those explored here. As such, I would encourage the authors to insert the word "possible" to say instead "... pointing to the POSSIBLE presence of ancient hybridizations between them".. since the results could also point to ILS, even if their preferred explanation is admixture. Note also that the authors have indeed discovered an allopolyploidy (=hybridization) event in their work, and significant discordance from MSCquartets might stem from this hybridization when quartet counts from within their larger gene trees are explored by the method. Thus, if it were me, I would couch my language on the preference for post-polyploid introgression accounting for the discordance - and my insertion of the word POSSIBLE above would do that I think... indeed, even the MSCquartets authors couch their own use of the word hybridization in their paper describing the method (note the word PERHAPS, below): "With exhaustive alternative hypotheses, rejection of the null indicates in the first case that the MSC on the specific species tree topology is rejected, and in the second that the MSC on any tree is rejected (PERHAPS due to hybridization).."

Response 1: Thanks for your suggestion. We have inserted the word "possible" in line 294.

Reviewer #4 (Remarks to the Author):

I agree with reviewer 2 that the rejection of the null hypothesis in MSCQuartets does not automatically imply hybridization as there are many sources of gene tree discordance that are not properly explained by the tree-based coalescent model. That is, the null hypothesis in MSCQuartets is that a coalescent model on a tree (that accounts solely for ILS as source of gene tree discordance) is enough to explain the gene tree discordance observed in the data. Rejecting this claim means that ILS is not enough to explain the observed gene tree discordance, yet we are unable to conclude with certainty that the alternative has to be hybridization. Combined with other analyses, the authors might have strong suspicion that hybridization is the cause for discordance, but this conclusion cannot be reached from MSCQuartets alone.

I echo the reviewer with the addition of the word "possible".

Response 2: Done

Minor comment:

- Line 277: "that does not require a species tree". MSCQuartets uses intrinsically a species tree to obtain the expected CFs under the coalescent model. Maybe I would change this sentence to "that does not require a pre-specified species tree".

Response 3: Done.